# FREQUENCY-AWARE TRANSFORMER FOR LEARNED IMAGE COMPRESSION

**Han Li**[1], **Shaohui Li**[2*], **Wenrui Dai**[1*], **Chenglin Li**[1], **Junni Zou**[1], **Hongkai Xiong**[1]

[1]Shanghai Jiao Tong University,
[2]Tsinghua Shenzhen International Graduate School, Tsinghua University
{qingshi9974,daiwenrui,lcl1985,zoujunni,xionghongkai}@sjtu.edu
lishaohui@sz.tsinghua.edu.cn

## ABSTRACT

Learned image compression (LIC) has gained traction as an effective solution for image storage and transmission in recent years. However, existing LIC methods are redundant in latent representation due to limitations in capturing anisotropic frequency components and preserving directional details. To overcome these challenges, we propose a novel frequency-aware transformer (FAT) block that for the first time achieves multiscale directional ananlysis for LIC. The FAT block comprises frequency-decomposition window attention (FDWA) modules to capture multiscale and directional frequency components of natural images. Additionally, we introduce frequency-modulation feed-forward network (FMFFN) to adaptively modulate different frequency components, improving rate-distortion performance. Furthermore, we present a transformer-based channel-wise autoregressive (T-CA) model that effectively exploits channel dependencies. Experiments show that our method achieves state-of-the-art rate-distortion performance compared to existing LIC methods, and evidently outperforms latest standardized codec VTM-12.1 by 14.5%, 15.1%, 13.0% in BD-rate on the Kodak, Tecnick, and CLIC datasets. Code will be releaset at `https://github.com/qingshi9974/ICLR2024-FTIC`

## 1 INTRODUCTION

Learned image compression (LIC) models have emerged as a promising solution to image storage and transmission and outperform traditional codecs in rate-distortion (R-D) metrics. Theoretically, LIC leverages the nonlinear transforms to alternatively enable a multi-dimensional quantizer with adaptive quantization cells, and consequently, exceeds the traditional transform coding schemes with restricted constructions. Early LIC models usually adopt convolutional neural networks (CNNs) to achieve nonlinear analysis and synthesis transforms (Ballé et al., 2018; Minnen et al., 2018). However, the local receptive fields of CNNs limit the representative ability and render redundant latent representations. Recent works (Zou et al., 2022; Zhu et al., 2022; Liu et al., 2023) employ attention modules or transformers to capture the non-local spatial relationship for better R-D performance.

Despite the success of transformers, there lacks an interpretation on the frequency characteristics of natural images in LIC models, which plays a crucial role in conventional image representation. For example, wavelet analysis allows the decomposition of an image into multiscale subbands, revealing details and structural information within different frequency ranges (Mallat, 1989). Consequential multiscale geometric analysis (MGA) tools further extend wavelet transforms with improved directional analysis realized by frequency partitioning (Candès & Donoho, 2000), space-frequency tiling (Donoho & Huo, 2002), and multi-directional filters (Do & Vetterli, 2005). The elegant mathematical foundations and wide applications of MGA also inspire us in exploring deep learning-based image representation with directional analysis.

In this paper, we propose a frequency-aware transformer for constructing the nonlinear transforms in LIC, which captures multiscale and directional frequency components of natural im-

---

*Corresponding authors: Wenrui Dai; Shaohui Li.

ages with isotropic and anisotropic window attention. Specifically, we first introduce frequency-decomposition window attention (FDWA), which comprises four types of attention modules that capture the low-frequency, high-frequency, vertical, and horizontal components. Subsequently, we develop a frequency-modulation feed-forward Network (FMFFN) to modulate the frequency components adaptively. Furthermore, we propose a transformer-based channel-wise autoregressive (T-CA) entropy model that exploits correlations across the directional components with causal masks.

To our best knowledge, this paper is the first to achieve transformer-based directional analysis in LIC. Different from handcrafted directional analysis tools in MGA, we achieve end-to-end optimization to realize novel anisotropic window attention. Distinguished from existing transformer-based image compression methods (Lu et al., 2022; Zhu et al., 2022; Liu et al., 2023; Zafari et al., 2023), we exploit the crucial directional information in nonlinear transforms to break the limitation of existing CNN-based and transformer-based LIC models. In summary, our contributions include:

- We propose a frequency-decomposition window attention (FDWA), which leverages diverse window shapes to capture frequency components of natural images to achieve more efficient latent representation in an end-to-end learned manner.

- We develop a frequency-modulation feed-forward network (FMFFN) that adaptively ensemble frequency components for improved R-D performance.

- We present a transformer-based channel-wise autoregressive model (T-CA) for effectively modeling dependencies across frequency components.

- Experiments show that our method achieves state-of-the-art R-D performance, and outperforms VTM-12.1 by 14.5%, and 15.1%, 13.0% in BD-rate on Kodak, Tecnick, and CLIC Professional Validation datasets respectively.

## 2 RELATED WORK

### 2.1 TRANSFORMER-BASED IMAGE COMPRESSION

Transformers (Vaswani et al., 2017) have achieved remarkable success in various computer vision tasks (Dosovitskiy et al., 2020; Carion et al., 2020; Li et al., 2023) due to its powerful non-local modeling ability. Recently, researchers have also incorporated transformer into learned image compression. Zhu et al. (2022) first demonstrate that nonlinear transforms built on Swin-Transformer (Liu et al., 2021) can achieve superior compression efficiency compared to those built on CNNs. Liu et al. (2023) combine transformers and CNNs to aggregate non-local and local information for more expressive nonlinear transforms. However, the standard window self-attention used in these works ignores the underlying different frequency components of natural image, which hinders the extraction of compact latent representations. To address this issue, our paper proposes a novel frequency-aware transformer (FAT) block.

### 2.2 AUTOREGRESSIVE ENTROPY MODELING

Entropy modeling is crucial for learned image compression models. A precise entropy model can eliminate the coding redundancy and minimize the size of compressed images. Most existing works are developed based on joint autoregression and hyperprior model (Minnen et al., 2018), in which the autoregression module captures the dependencies within the latent representations and reduces coding redundancy. The autoregression modules



Figure 1: Illustration of the proposed frequency-decomposition widow attention (FDWA) that realizes multiscale and directional decomposition. The first column shows diverse window shapes for capturing different frequency components, the second column visualizes the extracted features, and the third column presents the Fourier spectrum of the features.

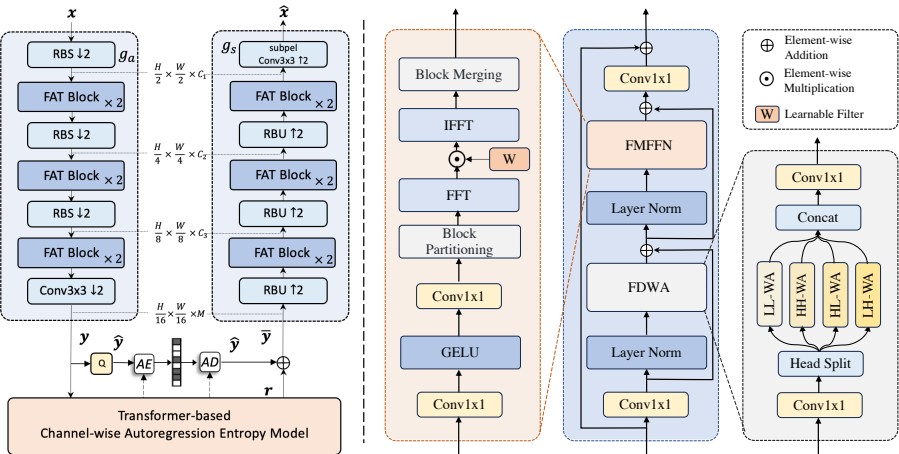

(a) Overall architecture of our *FAT-LIC*

(b) Frequency-Aware Transformer Block

Figure 2: Overview of the proposed Frequency-aware Transformer-based learned Image Compression (FTIC) model. Multiple residual blocks with stride (RBS), residual block Upsampling (RBU), and frequency-aware transformer (FAT) blocks are employed in building the nonlinear transforms (*i.e.*, analysis transform $g_a(\cdot)$ and synthesis transform $g_s(\cdot)$). For each two concatenated FAT blocks, the first one employs the regular frequency-decomposed window attention (FDWA) and the second one performs shift-window operations.

could be classified as spatial autoregression (Minnen et al., 2018) (SA), channel-wise autoregression (Minnen & Singh, 2020) (CA), and their combination (He et al., 2022). Recent works employ transformers to capture long-range dependency and improve the preciseness of entropy modeling. Qian et al. (2022) utilize global self-attention to capture long-range spatial dependency in distribution estimation. Koyuncu et al. (2022) propose a spatial-channel attention to fully exploit latent dependency and improve R-D performance. However, the heavy memory usage and dramatic computational complexity render these methods impractical for real-world image compression, especially for high-resolution images. Liu et al. (2023) incorporate Swin-Transformer into channel-wise autoregressive entropy model to capture additional spatial dependency. However, the swin-transformer increases the model size while the R-D improvement over the anchor model seems insignificant. Different from these works, the proposed T-CA concentrates on improving channel-wise attention without obviously increasing additional parameters.

## 2.3 FREQUENCY DECOMPOSITION IN LEARNED IMAGE COMPRESSION

Traditional image codecs employ subband decomposition to decorrelate frequency components of images to achieve compact representations. Recent learned image compression models also benefit from the explicit frequency decomposition. Ma et al. (2019; 2020) introduce wavelet-like transform in LIC, but it is restricted by the lifting scheme, limiting the representation ability of network and constraining the latent space. Gao et al. (2021) propose a frequency decomposition model that manipulates the low- and high-frequency components in the input image separately. Zafari et al. (2023) utilize HiLo attention (Pan et al., 2022) to disentangle low- and high-frequency components. However, the global self-attention in their approach poses computational challenges when dealing with large input images and directional decomposition can not be achieved. In this work, we propose FDWA with diverse window size to capture the multiscale and directional frequency components by transformer simultaneously.

## 3 METHODS

### 3.1 OVERVIEW

Figure 2 illustrates the architecture of the proposed **F**requency-aware **T**ransformer-based learned **I**mage **C**ompression (FTIC) model. Given a raw image $x$, the analysis transform $g_a(\cdot)$ maps it to a latent representation $y$. Then, quantization operator $Q(\cdot)$ discretizes $y$ to $\hat{y}$. $\hat{y}$ is subsequently

losslessly encoded by a range coder. Here, we assume that $\boldsymbol{y}$ follows a Guassian distribution of which the parameters (*i.e.*, mean $\boldsymbol{\mu}$ and scale $\boldsymbol{\sigma}$) are predicted by the proposed transformer-based channel-wise autoregressive (T-CA) entropy model, as presented in Figure 3. Following the settings in the original channel-wise autoregressive entropy model (Minnen & Singh, 2020), we divide $\boldsymbol{y}$ into $n_s$ even slices $\{\boldsymbol{y}_1, \boldsymbol{y}_2, ..., \boldsymbol{y}_{n_s}\}$ and feed these slices to T-CA model. Therefore, the encoded slices can provide powerful contextual information when encoding subsequent slices. Formally, the forward process of T-CA model holds as below. The hyperprior $\phi$ is obtained via a pair of hyper-encoder $h_a(\cdot)$ and hyper-decoder $h_s(\cdot)$:

$$\boldsymbol{z} = h_a\left(\boldsymbol{y}\right), \ \hat{\boldsymbol{z}} = Q(\boldsymbol{z}), \ \boldsymbol{\phi} = h_s(\hat{\boldsymbol{z}}). \tag{1}$$

Moreover, the T-CA entropy model outputs the estimated parameters of the Gaussian distributions:

$$\boldsymbol{r}_i, \boldsymbol{\mu}_i, \boldsymbol{\sigma}_i = \text{T-CA}(\boldsymbol{\phi}, \hat{\boldsymbol{y}}_{<i}), \ 1 \leq i < n_s, \tag{2}$$

where $\boldsymbol{\mu}_i$ and $\boldsymbol{\sigma}_i$ are mean and scale values of Gaussian distributions, and $\boldsymbol{r}_i$ is the predicted latent residual that reduces quantization errors of $\boldsymbol{y}_i$. A refined latent $\bar{\boldsymbol{y}}$ could be obtained by summing up the latent $\hat{\boldsymbol{y}}$ and latent residual $\boldsymbol{r}$, *i.e.*, $\bar{\boldsymbol{y}} = \hat{\boldsymbol{y}} + \boldsymbol{r}$ with $\boldsymbol{r} = \text{Concat}(\boldsymbol{r}_1, \boldsymbol{r}_2, ..., \boldsymbol{r}_{n_s})$. Consequently, the final $\hat{\boldsymbol{x}}$ is reconstructed by feeding refined latent $\bar{\boldsymbol{y}}$ to the synthesis transform $g_s(\cdot)$, *i.e.*, $\hat{\boldsymbol{x}} = g_s(\bar{\boldsymbol{y}})$.

To train our *FTIC* model, we formulate the problem as a Lagrangian multiplier-based R-D optimization, in which the loss function is defined as:

$$\mathcal{L} = \mathcal{R}(\hat{\boldsymbol{y}}) + \mathcal{R}(\hat{\boldsymbol{z}}) + \lambda \cdot \mathcal{D}(\boldsymbol{x}, \hat{\boldsymbol{x}}) \tag{3}$$

where a Lagrangian multiplier $\lambda$ controls the trade-off between rate and distortion. Different $\lambda$ values are corresponding to different bitrates. $\mathcal{D}(\boldsymbol{x}, \hat{\boldsymbol{x}})$ denotes the distortion term between the raw image $\boldsymbol{x}$ and reconstructed image $\hat{\boldsymbol{x}}$. $\mathcal{R}(\hat{\boldsymbol{y}}), \mathcal{R}(\hat{\boldsymbol{z}})$ denote the bitrates of latents $\hat{\boldsymbol{y}}$ and $\hat{\boldsymbol{z}}$. Please refer to Appendix A for the detailed model architecture.

## 3.2 FREQUENCY-AWARE TRANSFORMER BLOCK

We aim at building efficient frequency decomposition within the end-to-end optimized image compression framework. To this end, we propose a novel frequency-aware transformer (FAT) block, which achieves naïve multiscale and directional decomposition. Specifically, the FAT block exploits frequency-decomposed window attention (FDWA) mechanism, which decomposes the input image into four components (*i.e.*, low-frequency, high-frequency, vertical, and horizontal components presented in Figure 1). Then a frequency-modulation feed-forward network (FMFFN) modulates the decomposed components and eliminate the potential redundancy across frequency components. The following subsections elaborates the implementations of FDWA and FMFFN.

### 3.2.1 FREQUENCY-DECOMPOSED WINDOW ATTENTION

Recent studies (Park & Kim, 2022; Pan et al., 2022) have demonstrated that a typical self-attention is in fact a low-pass filter, and local window attention with a smaller window size captures fine-grained high-frequency information. Therefore, leveraging windows of varying sizes enables the extraction of multiscale frequency components. However, the square windows employed in existing self-attention are inefficient for capturing directional frequency information due to their isotropic characteristic.

To eliminate the limitation on directional frequency decomposition, we propose a frequency-decomposed window attention (FDWA) module, which performs self-attention with four diversely shaped windows in parallel. In our experiments, the sizes (height $\times$ width) of these four windows are $4s \times 4s$, $s \times s$, $s \times 4s$, and $4s \times s$, with $s$ being the basic window size. These windows correspond to the low-frequency, high-frequency, vertical, and horizontal components. From the perspective of separable two-dimensional frequency decomposition, such implementation equals to a permutation and combination of low(L)- and high(H)-frequency decomposition on each dimensions. Therefore, we also term the window attention (WA) achieved by these windows as LL-WA, HH-WA, HL-WA, and LH-WA.

Given the input hidden representation $\boldsymbol{X} \in \mathbb{R}^{C \times H \times W}$, where $H \times W$ is the spatial resolution and $C$ denotes the number of channels, we first linearly project it into $K$ heads. Then we split the $K$

heads evenly into four parallel groups with $K/4$ heads in each group (assuming $K$ is a multiple of 4), and each group performs a specific type of self-attention. Without loss of generality, we use the case of LH-WA as an example.

*Example: LH-WA.* $\boldsymbol{X}$ is evenly partitioned into non-overlapping windows $[\boldsymbol{X}^1, \cdots, \boldsymbol{X}^M]$, where each window contains $4s \times s$ tokens. Assuming the projected queries, keys, and values of the $k$th ($\frac{3}{4}K < k \leq K$) head are $d_k$-dimensional tensors, then output of the LH-WA for the $k$th head can be obtained as follows.

$$
\begin{aligned}
\boldsymbol{X} &= [\boldsymbol{X}^1, \cdots, \boldsymbol{X}^M], \\
\boldsymbol{Y}_k^i &= \text{Attention}(\boldsymbol{X}^i \boldsymbol{W}_k^Q, \boldsymbol{X}^i \boldsymbol{W}_k^K, \boldsymbol{X}^i \boldsymbol{W}_k^V), \\
\text{LH-WA}_k(\boldsymbol{X}) &= [\boldsymbol{Y}_k^1, \boldsymbol{Y}_k^2, \cdots, \boldsymbol{Y}_k^M],
\end{aligned}
\tag{4}
$$

where $\boldsymbol{X}^i \in \mathbb{R}^{C \times 4s \times s}$, and $M = \frac{H \times W}{4s \times s}$, $i = 1, \cdots, M$. $\boldsymbol{W}_k^Q, \boldsymbol{W}_k^K, \boldsymbol{W}_k^V \in \mathbb{R}^{C \times d_k}$ are the projection matrices of queries, keys and values for the $k$th head, respectively, with $d_k = C/K$.

The process for other three types of window attention can be derived easily, and their output for the $k$th head are denoted as $\text{LL-WA}_k(\boldsymbol{X}), \text{HH-WA}_k(\boldsymbol{X}), \text{HL-WA}_k(\boldsymbol{X})$, respectively. Finally the output of these four parallel groups will be concatenated to form the overall output:

$$
\begin{aligned}
\text{FDWA}(\boldsymbol{X}) &= \text{Concat}[\text{head}_1, \cdots, \text{head}_K]\boldsymbol{W}^O, \\
\text{with } \text{head}_k &= \begin{cases} \text{LL-WA}_k(\boldsymbol{X}) & k = 1, \cdots, K/4 \\ \text{HH-WA}_k(\boldsymbol{X}) & k = K/4 + 1, \cdots, K/2 \\ \text{HL-WA}_k(\boldsymbol{X}) & k = K/2 + 1, \cdots, 3K/4 \\ \text{LH-WA}_k(\boldsymbol{X}) & k = 3K/4 + 1, \cdots, K \end{cases}
\end{aligned}
\tag{5}
$$

where $\boldsymbol{W}^O \in \mathbb{R}^{C \times C}$ is the projection matrix that implements interaction between different frequency components.

### 3.2.2 Frequency-Modulation Feed-forward Network

The feed-forward network (FFN) in transformer is employed to refine the features produced by self-attention. As described above, FDWA produces features with diverse frequency components. However, the contributions of these frequency components in compression are not equal. For instance, high-bitrate models may require more high-frequency components to recover edges and fine-grained details, while low-bitrate models mainly rely on low-frequency components to reconstruct the overall structure.

To alternatively decide the frequency components and further eliminate the redundancy across different components, we develop a Frequency-Modulation FFN (FMFFN) that can adaptively modulate frequency components. Specifically, we apply a block-based fast Fourier transform (FFT) to transform the feature $\boldsymbol{X}_{\text{ffn}}$ obtained by a standard FFN into frequency domain. Next, we introduce a learnable filter matrix $\boldsymbol{W}$ to alternatively suppress or amplify all frequency components by element-wise multiplication in the frequency domain, obtaining the frequency-modulated feature $\boldsymbol{X}_{\text{fm}}$. Subsequently, $\boldsymbol{X}_{\text{fm}}$ is inversely transformed using the inverse FFT (IFFT) and reshaped to obtain the refined feature $\boldsymbol{X}_{\text{out}}$. The overall process of FMFFN can be formulated as follows:

$$
\begin{aligned}
\boldsymbol{X}_{\text{ffn}} &= \text{Conv}_{1 \times 1}(\text{GELU}(\text{Conv}_{1 \times 1}(\boldsymbol{X}))) \\
\boldsymbol{X}_{\text{fm}} &= \mathcal{F}\left[(\mathcal{B}(\boldsymbol{X}_{\text{ffn}})] \odot W \right. \\
\boldsymbol{X}_{\text{out}} &= \mathcal{B}^{-1}\left[(\mathcal{F}^{-1}(\boldsymbol{X}_{\text{fm}})]\right.
\end{aligned}
\tag{6}
$$

where $\boldsymbol{X}$ is the input hidden representation of the FMFFN, $\odot$ denotes the element-wise multiplication. $\mathcal{F}(\cdot)$ and $\mathcal{F}^{-1}(\cdot)$ denote the FFT and in IFFT, respectively. $\mathcal{B}(\cdot)$ and $\mathcal{B}^{-1}(\cdot)$ denote block partitioning and block merging operation, respectively. The block size is set to $4s \times 4s$, which corresponds to the maximum window size in our FDWA. By performing block partitioning, the size of the weight matrix $\boldsymbol{W}$ becomes independent of the input image size. This characteristic allows our FMFFN to generalize to input images of arbitrary sizes.

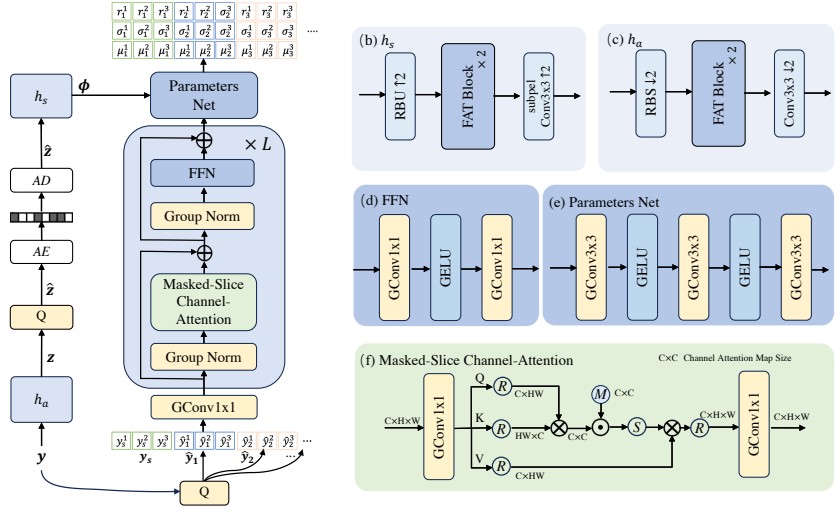

Figure 3: Proposed Transformer-based Channel-wise Autoregressive (T-CA) entropy model, the hyperprior path is also included. For briefness, we suppose each slice has 3 channels in (a). GConv $n \times n$ denotes the group convolutions with kernel size of $n \times n$.

### 3.3 TRANSFORMER-BASED CHANNEL-WISE AUTOREGRESSIVE (T-CA) ENTROPY MODEL

Previous channel-wise autoregressive models (Minnen & Singh, 2020; He et al., 2022) divide latent $\boldsymbol{y}$ to several slices along the channel dimension, and then utilize separate CNNs to establish the dependencies of each slice on previously decoded slices. However, the coefficient distribution in each channels vary dramatically across different input images, so that the fixed weights in CNNs can not fully exploit the channel-wise correlations. In this paper, we propose a novel transformer-based channel-wise autoregressive (T-CA) entropy model as illustrated in Figure 3. By introducing channel-attention in channel-wise autoregression, our T-CA can effectively capture the inter- and intra-slices channel dependencies, leading to more precise distribution estimation.

Our T-CA has $L$ transformer layers with a similar architecture of vision transformer (ViT) (Dosovitskiy et al., 2020), while we utilize channel attention instead of spatial attention. For the input quantized latent representation $\hat{\boldsymbol{y}} \in \mathbb{R}^{M \times H \times W}$ where $M, H, W$ denote the number of channels, height and the width, we first evenly divide it into $n_s$ slices $\{\hat{\boldsymbol{y}}_1, \hat{\boldsymbol{y}}_2, \cdots, \hat{\boldsymbol{y}}_{n_s}\}$, so that each slice has $M_s$ channels (*i.e.*, $\hat{\boldsymbol{y}}_i = \{\hat{y}_i^1, \hat{y}_i^2, \cdots, \hat{y}_i^{M_s}\}$, where $i$ is the index of slice and $M_s = M/n_s$ ). Before feeding the slices to the transformer, we project each slice to $(r \cdot M_s)$ channels independently, where $r$ is a projection ratio larger than 1. This process can be simply implemented by inputting $\hat{\boldsymbol{y}}$ to a group convolution, of which the input and output channels are $M$ and $(r \cdot M)$, kernel size is $1 \times 1$ and the number of group $n_g$ is equal to $n_s$.

We modify the standard transformer layer to guarantee the causality of the entropy coding. First, we introduce the masked-slice channel attention by incorporating a causal slice-wise mask, which ensures that the slices not yet encoded do not affect other slices. Besides, we also employ a pseudo start slice $\boldsymbol{y}_s$ similar to Mentzer et al. (2022). Second, we replace LayerNorm with GroupNorm and substitute all linear layers with $1 \times 1$ group convolution layers, where the number of groups $n_g = n_s$. These group-wise operations not only facilitate improved modeling of intra-slice channel dependencies, but also reduce the model complexity, leading to improved computational efficiency. Moreover, we utilize group convolutions to predict the Guassian parameters $\mu, \sigma$ and the latent residual $r$ from the contextual information and hyperprior. Please refer to Appendix-A.3 for details.

## 4 EXPERIMENTS

### 4.1 EXPERIMENTAL SETUP

We train the proposed FTIC models on the *Flickr2W* (Liu et al., 2020) and ImageNet-1k (Deng et al., 2009) dataset for 3.2M steps with a batch size of 8. The model is optimized using Adam optimizer

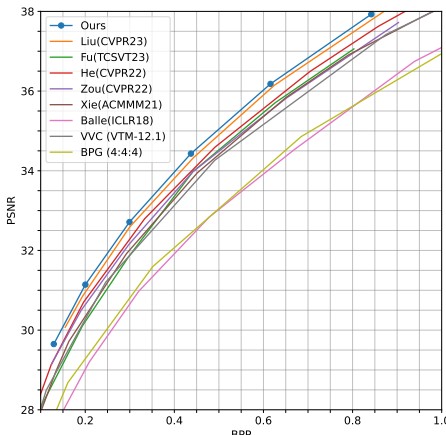 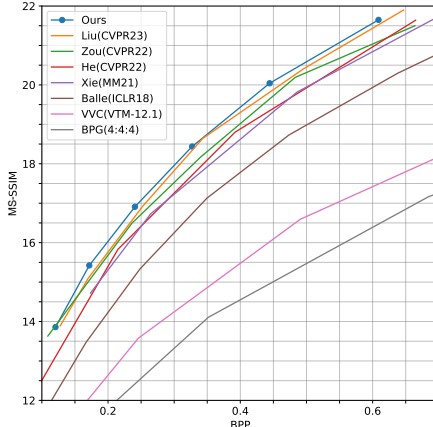

Figure 4: R-D performance evaluated on the Kodak dataset. The compared methods include state-of-the-art LIC models and handcrafted image codecs. Left: PSNR; right: MS-SSIM.

with the learning rate initialized as 1e-4. Specifically, we follow Zhu et al. (2022) to adopt the multi-stage training strategy. The details can be found in Appendix-B. We ues R-D loss in Equation 3 to optimize the model. Two kinds of quality metrics, *i.e.*, mean square error (MSE) and multiscale structural similarity (MS-SSIM), are used to measure the distortion $\mathcal{D}$. The Lagrangian multiplier used for training MSE-optimized models are $\{0.0025, 0.0035, 0.0067, 0.0130, 0.0250, 0.0483\}$, and those for MS-SSIM-optimized models are $\{2.40, 4.58, 8.73, 16.64, 31.73, 60.50\}$.

For FAT blocks, the base window size $s$ is set as 4 in the nonlinear transforms (*i.e.*, $g_a(\cdot)$ and $g_s(\cdot)$), and set as 1 in the hyperprior transforms ( *i.e.*, $h_a(\cdot)$ and $h_s(\cdot)$). The channel number $M$ of the latent $\boldsymbol{y}$ is set as 320, while the channel number $N$ of hyper latent $\boldsymbol{z}$ is set as 192. For T-CA entropy model, the number of slices $n_s$ is set as 5, the number of transformer layers $L$ is set as 12, and projection ratio $r$ is set as 4. More hyper parameters of architecture can be found in Appendix-A.1. We use NVIDIA GeForce RTX 4090 and Intel Xeon Platinum 8260 to conduct the following experiments.

We evaluate our the proposed model on three benchmark datasets, *i.e.*, Kodak image set (Kodak, 1993) with 24 images of $768 \times 512$ pixels, Tecnick testset (Asuni & Giachetti, 2014) with 100 images of $1200 \times 1200$ pixels, CLIC Professional Validation dataset (CLIC, 2021) with 41 images of at most 2K resolution. We use both PSNR and MS-SSIM to measure the distortion, while bits per pixel (BPP) is used to evaluate bitrates.

## 4.2 RATE-DISTORTION PERFORMANCE

We compare our method with the state-of-the-art (SOTA) methods including the traditional image codecs BPG and VTM-12.1, and recent LIC models (Ballé et al., 2018; Cheng et al., 2020; Xie et al., 2021; He et al., 2022; Zou et al., 2022; Liu et al., 2023; Fu et al., 2023). The R-D performance on Kodak dataset is shown in Figure 4. We use both PSNR and MS-SSIM as quality metric to evaluate the performance of our method. The results of Tecnick and CLIC datasets are shown in Figure 10 in the Appendix, respectively. These additional results demonstrate that our method consistently achieves excellent performance across all three datasets. Furthermore, we report the BD-rate (Bjontegaard, 2001) results to quantify the average bitrate savings with equal reconstruction quality, with VTM-12.1 as the anchor. Our method achives the state-of-the-art performance and outperforms VTM-12.1 by 14.5%, 15.1%, 13.0% in BD-rate on the Kodak, Tecnick, and CLIC dataset, respectively.

## 4.3 ABLATION STUDIES AND ANALYSIS

**Effect of the architecture of FAT block.** We conduct ablation studies to investigate the effectiveness of the FAT block. For the baseline model, we substitute our proposed FDWA with the

Table 1: Comparison on architectures of FAT Block, evaluated on the Kodak dataset. The GFLOPs of running analysis transform (*i.e.*, $g_a(\cdot)$) and the overall model parameters are listed for complexity comparison. The BD-rate is presented for performance comparison with VTM-12.1 as the anchor.

| Methods | Larger Window | FDWA | FMFFN | GFLOPs | #Params | BD-rate |
|---|---|---|---|---|---|---|
| Baseline | | | | 77.3 | 70.29M | -9.3% |
| Variant 1 | ✓ | | | 91.3 | 70.50M | -11.2% |
| Variant 2 | | ✓ | | 82.2 | 70.36M | -13.3% |
| Variant 3 (FAT Block) | | ✓ | ✓ | 83.1 | 70.97M | -14.5% |
| VTM-12.1 | - | - | - | - | - | 0% |

standard window-based self-attention using a window size of 4×4, and replaced our FMFFN with a conventional FFN. As shown Table 1, adopting a larger window size of 16× 16 can improve performance. However, this also introduces higher computational complexity. Furthermore, the adoption of FDWA results in significant improvement while maintaining comparable complexity. This indicates that the performance gains of introducing FDWA are derived from the ability to extract diverse frequency components rather than solely relying on a large receptive field. Finally, by further introducing FMFFN, we achieve the state-of-the-art BD-rate with only slightly increased computational complexity and model parameters.

**Ablation study on T-CA.** We then evaluate the effect of our proposed T-CA entropy model. Here, we adopt the CNN-based nonlinear transforms of Minnen et al. (2018) as $g_a(\cdot)$ and $g_s(\cdot)$ for convenient comparison. Specifically, we train all the models by loading the pretrained transforms weights of *mbt2018-mean* model from CompressAI library (Bégaint et al., 2020) and fine-tune them for 1M batches. The BD-rate is evaluated on the Kodak dataset with BPG as the anchor. In addition, we provide the number of parameters of the entropy models to show their complexity.

We first compare our T-CA entropy model with CHARM (Minnen & Singh, 2020) to show the powerful ability of modeling channel dependency through channel attention. As shown in Table 2, our T-CA can outperform CHARM significantly with the same number of slices (*i.e.*, 5). Furthermore, it achieves superior R-D performance over CHARM with half the number of slices (5 vs 10) and reduced model complexity.

Table 3 reports how different parameters impact the R-D performance and the complexity of our T-CA entropy model. The results show that, enlarging the number of transformer layers from 4 to 12 can boost the performance, but using transformer layers more

Table 2: Comparison of the proposed T-CA with CHARM.

| Model | #Params | BD-rate |
|---|---|---|
| CHARM ($n_s$=5) | 18.3M | -14.2% |
| CHARM ($n_s$=10) | 34.8M | -15.9% |
| T-CA ($n_s$=5) | 30.4M | -19.2% |
| BPG | - | 0% |

Table 3: Ablations on different parameters of T-CA. $L$ is the number of transformer layers of T-CA. $n_s$ is the number of slices.

| $L$ | $n_s$ | #Params | BD-rate |
|---|---|---|---|
| 4 | 5 | 14.5M | -17.6% |
| 8 | 5 | 22.4M | -18.3% |
| **12** | 5 | 30.4M | **-19.2%** |
| 16 | 5 | 38.4M | -19.0% |
| 12 | 4 | 37.8M | -19.0% |
| 12 | **5** | 30.4M | **-19.2%** |
| 12 | 8 | 19.3M | -18.0% |
| 12 | 10 | 15.6M | -18.4% |
| BPG | - | | 0% |

than 12 cannot bring additional benefits. In addition, we observe that increasing the number of slices $n_s$ does not lead to consistent performance improvement as demonstrated in Minnen & Singh (2020). This is due to that our T-CA model utilizes a shared transformer to establish channel dependency among different slices, and the number of parameters does not increase linearly with $n_s$ as CHARM (Minnen & Singh, 2020). Conversely, increasing $n_s$ in T-CA results in a reduction in number of parameters of the group convolution as well as the whole entropy model. Therefore, we select $n_s$ as 5 and $L$ as 12 to balance the model size and performance.

**Spectrum analysis of FDWA.** In Figure 5, we visualize the intensity of frequency component by applying Fast Fourier Transform (FFT) to the output feature maps from different attention modules in our FDWA. The visualisation clearly demonstrates that HH-WA captures more high frequencies,

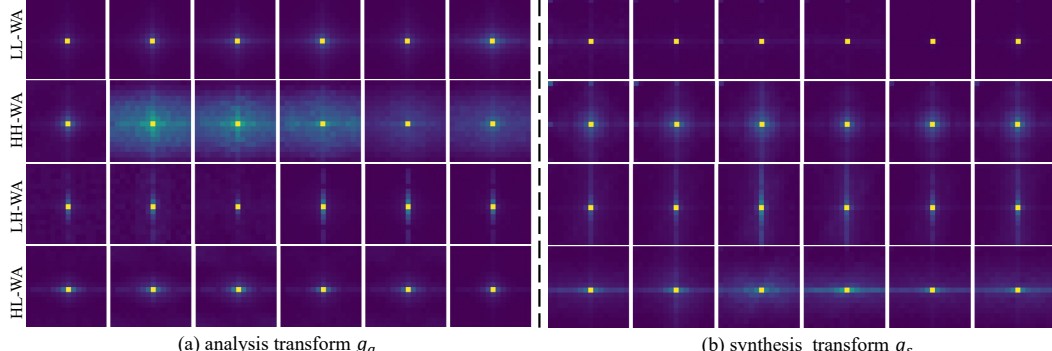

(a) analysis transform $g_a$        (b) synthesis transform $g_s$

Figure 5: Frequency intensity (16 × 16) from the output of FDWA at the last FAT block for both (a) analysis transform $g_a(\cdot)$ and (b) synthesis transform $g_s(\cdot)$. The model is trained with $\lambda$ set as 0.0483 and MSE as metric. We show 6 output channels for each of LL-WA, HH-WA, HH-WA, and LH-WA. The magnitude values are averaged over 100 samples. Lighter colors indicate larger magnitudes, while pixels closer to the center represent lower frequencies.

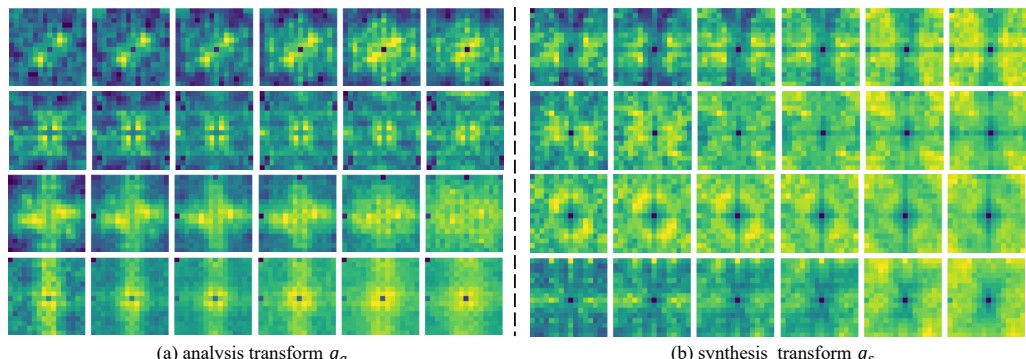

(a) analysis transform $g_a$        (b) synthesis transform $g_s$

Figure 6: Visualization of the frequency-domain learned filters of FMFFN at the deepest FAT block for both (a) analysis transform $g_a(\cdot)$ and (b) synthesis transform $g_s(\cdot)$. The four rows represent learned filters corresponding to four different channels, and the six columns from left to right represent the models with increasing bitrates.

while LL-WA mainly focuses on low frequencies. Additionally, HL-WA and LH-WA exhibit the ability to capture directional frequency. This observation strongly supports our primary goal of decomposing different frequency components within feature maps at a single attention layer.

**Visualization analysis of FMFFN.** In Figure 6, we follow Rao et al. (2021) to visualize the frequency domain learned filters $W$ in our FMFFN. For each row, from left to right, the six columns represent one specific filter of models with increasing bitrates. We can see that higher bitrates model contains more high frequency component, this findings also supports our motivation of introducing FMFFN. In this way, model can adaptively modulate different frequency components to achieve better R-D performance.

## 5 CONCLUSION

This paper proposes a novel approach for learned image compression (LIC) from the perspective of frequency decomposition. We address the challenge of modeling diverse frequency information in LIC by the proposed frequency-decomposition window attention (FDWA), which captures different orientation and spatial frequency components using various window sizes. We also introduce a frequency-modulation feed-forward network (FMFFN) module to adaptively amplify or suppress different frequency components for improved rate-distortion tradeoff. Furthermore, a transformer-based channel-wise autoregressive (T-CA) entropy model is developed to effectively learn channel dependencies. Experimental results demonstrate that the proposed method achieves state-of-the-art rate-distortion performance on commonly used datasets.

ACKNOWLEDGEMENT

This work was supported in part by the National Natural Science Foundation of China under Grant 62125109, Grant 62250055, Grant 61931023, Grant 61932022, Grant 62371288, Grant 62320106003, Grant 62301299, Grant T2122024, and Grant 62120106007.

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

# A MODEL ARCHITECTURE

## A.1 DETAILS OF OVERALL FRAMEWORK

Figure 7 shows the overall framework of our *FTIC* and detailed architecture of the used RBS and RBU. The number of channels $(C_1, C_2, C_3, M) = (96, 144, 256, 320)$. In both $g_a(\cdot)$ and $g_s(\cdot)$, the numbers of attention heads are configured as $(8, 8, 16, 16, 32, 32)$ from shallow to deep, whereas in the $g_a(\cdot)$ and $g_s(\cdot)$ all FAT blocks have 32 attention heads. The number of hyper latent channels is 192 for all layers, except for the output of $h_s$, which has 640 channels. For the T-CA entropy model, the number of slices $n_s$ is set to 5, the projection ratio $r$ is 4, and the number of attention heads in the masked-slice channel attention is 16.

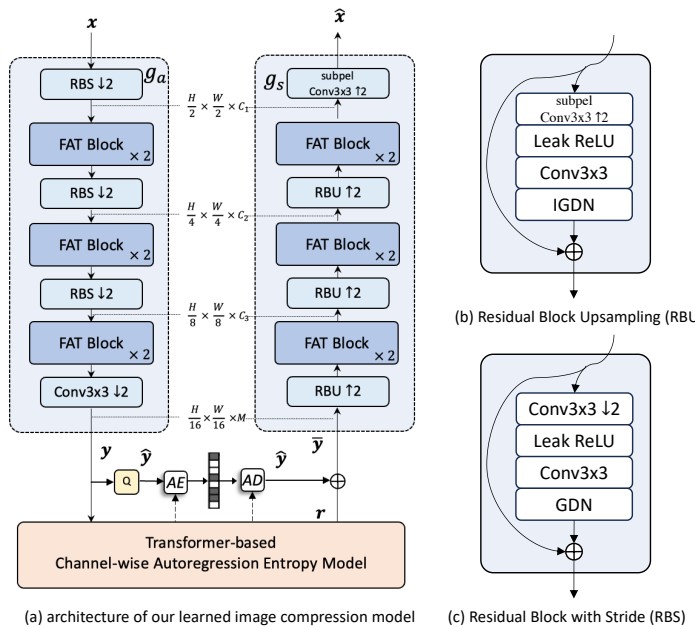

Figure 7: Left :The overall framework of our *FTIC*. Right: The architectures of RBS and RBU, which are firstly adopted in Cheng et al. (2020)

## A.2 DETAILS OF SHIFT-WINDOW OPERATION

In our *FTIC*, for each two concatenated FAT blocks, the first one employs the regular frequency-decomposed window attention (FDWA) and the second one performs shift-window operations. In the shift-window operation, the window size are shifted with a bias of the half height and width size. The exact operations for each type of window attention in our FDWA are illustrated in Figure 9.

## A.3 DETAILS OF ENTROPY PARAMETERS NETWORK

As shown in Figure 2, we utilize entropy parameters network to predict the mean, scale, and latent residual for each slice. To obtain the input of entropy parameters network, we first reshape the decoded hyperprior $\phi$ from $\mathbb{R}^{2M \times H \times W}$ to $\mathbb{R}^{M \times 2 \times H \times W}$ and reshape the output of fi-

Table 4: Detailed architectures of entropy parameters network.

| Entropy Parameters Network |
| --- |
| out: $(3 \times 320, 16, 16)$ |
| GConv $(3 \times 320, 3 \times 320, 3, 1, 1, 5)$ |
| GELU() |
| GConv $(3 * 320, 3 \times 320, 3, 1, 1, 5)$ |
| GELU() |
| GConv $(6 \times 320, 3 \times 320, 3, 1, 1, 5)$ |
| input: $(6 \times 320, 16, 16)$ |

nal transformer layer $y_{out}$ from $\mathbb{R}^{4M \times H \times W}$ to $\mathbb{R}^{M \times 4 \times H \times C}$. Subsequently, we concatenate these reshaped results, resulting in $y_{concat}$ reshaped from $\mathbb{R}^{M \times 6 \times H \times W}$ to $\mathbb{R}^{6M \times H \times W}$, which serves as the input for the entropy parameters network. This procedure ensures the causality by avoiding in-

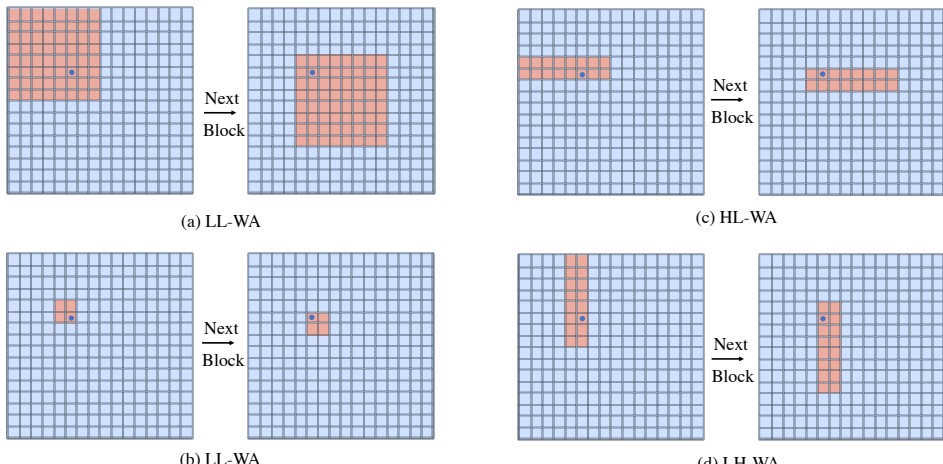

Figure 9: The shift-window operations for each type of attention in our FDWA.

(a) LL-WA

(b) LL-WA

(c) HL-WA

(d) LH-WA

formation interactions between different slices. The detailed architectures is presented in Table 4, where GConv denotes group convolution and the six parameters are input channels, output channels, kernel size, stride, padding, and the number of group.

# B TRAINING

Our learned image compression models are trained on Flickr2W and ImageNet-1k dataset. Previous works (He et al., 2022; Liu et al., 2023) encode each $\lceil y - \mu \rfloor$ to the bitstream instead of $\lceil y \rfloor$ and restore the coding-symbol as $\lceil y - \mu \rfloor + \mu$, which can significantly benefit the single Gaussian entropy model. However, in our autoregression model, this poses a train-test mismatch problem because we do not know the value of $\mu$ before passing through the model. To avoid this problem, in our methods, we adopt a mixed training strategy as follows:

In the first stage, we train our models with only a hyperpiror model as the entropy model to obtain a strong nonlinear transforms. The network architecture used in the first stage is presented in Figure 8. During this stage, we encode each $\lceil y - \mu \rfloor$ to the bitstream and restore the coding symbols as $\lceil y - \mu \rfloor + \mu$.

In the second stage, we load the transform weights (i.e., $g_a$ and $g_s$) from the checkpoint of pretrained first stage model and fine-tune the transforms together with the random initialized T-CA. In this stage, due to the causality limitation, we directly encode each $\lceil y \rfloor$ to the bitstream and restore the coding-symbol as $\lceil y \rfloor$.

The first stage is trained for 2M steps with a learning rate of 1e-4. Each batch contains 8 patches with the size of $256 \times 256$ randomly cropped from the training images. The second stage is trained for 1M steps with the same learning rate. Finally, we train the model with learning rate of 1e-5 for 200K steps using a larger crop size of $384 \times 384$. For all training, Adam optimizer is used without weighted decay.

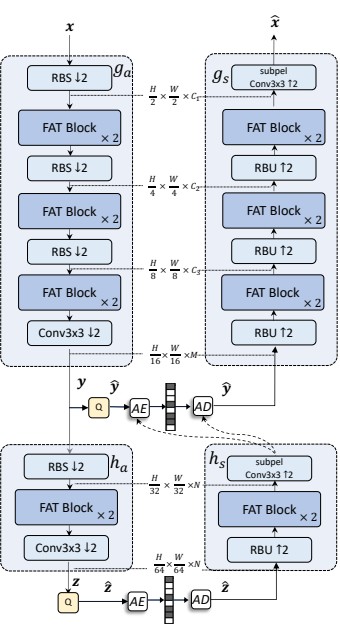

Figure 8: The framework of our *FTIC* with only hyperprior model as entropy model

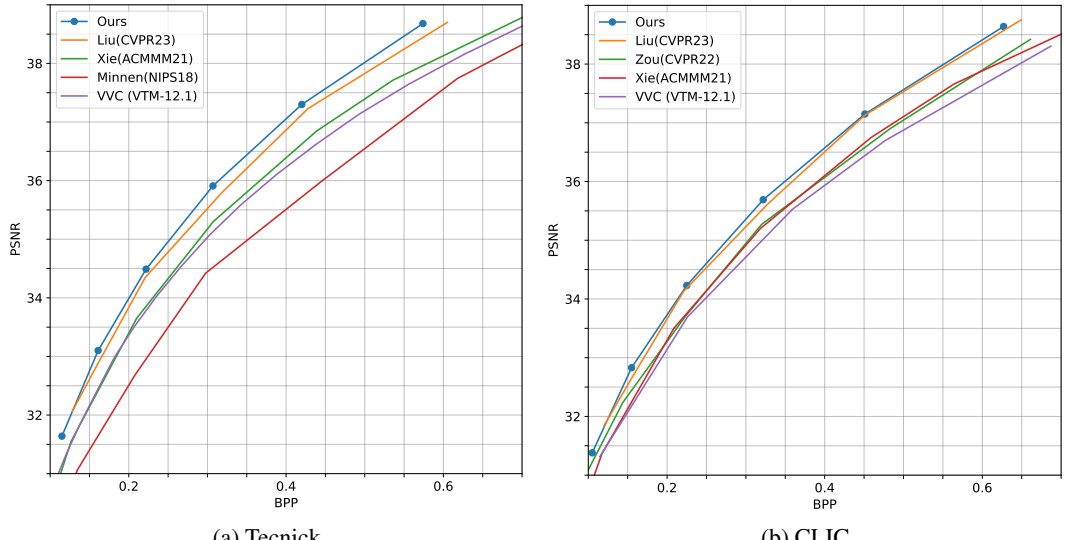

(a) Tecnick            (b) CLIC

Figure 10: R-D performance evaluated on (a) the Tecnick dataset and (b) the CLIC Professional Validation dataset.

Table 5: Comparison on coding complexity evaluated on the Kodak dataset. The BD-rate is presented for R-D performance comparison with VTM-12.1 as the anchor.

| Model | GFLOPs | | Inference Latentcy (ms) | | #Params | BD-rate |
|---|---|---|---|---|---|---|
| | Enc. | Dec. | Enc. | Dec. | | |
| Cheng et al. (2020) | 154 | 229 | >1000 | >1000 | 26.60M | 3.6% |
| Minnen & Singh (2020) | 101 | 100 | 56 | 43 | 55.13M | 1.1% |
| Zhu et al. (2022) | 116 | 116 | 110 | 99 | 32.71M | -3.3% |
| Zou et al. (2022) | 285 | 276 | 97 | 101 | 99.58M | -4.3% |
| Liu et al. (2023) | 317 | 453 | 255 | 322 | 76.57M | -11.9% |
| Ours | 141 | 349 | 125 | 242 | 70.97M | -14.5% |
| VTM-12.1 | - | - | - | - | - | 0% |

## C  ADDITIONAL EXPERIMENTAL RESULTS

### C.1  R-D PERFORMANCE ON TECNICK AND CLIC PROFESSIONAL VALIDATION DATESETS

We provide the additional rate-distortion results on Tecnick and CLIC Professional Validation datesets in Figure 10. The proposed model achieves state-of-the-art on both datasets.

### C.2  COMPARISON ON CODING COMPLEXITY

We compare the coding complexity of the proposed FTIC with existing state-of-the-art LIC models in Table 5. The coding complexity is measured by inference latency during encoding and decoding process. The experiments are conducted on a single NVIDIA GeForce RTX 4090 with 24 GB memory. Cheng et al. (2020) exhibits a high inference time due to the spatial auto-regressive entropy model. The experiments show that the proposed method could achieve the best BD-rate reduction with a tolerable coding complexity.

### C.3  COMPARISON ON TRAINING SPEED AND GPU MEMORY REQUIREMENT

We compare the training speed and GPU memory requirement of the proposed FTIC with existing state-of-the-art LIC models in Table 6. Compared with existing transformer-based LIC methods, the proposed method exhibits comparable training speed and GPU memory requirement.

Table 6: Comparison on training speed and gpu memory requirement. For training, each batch contains 8 images with resolution of 256×256. For test, each batch contains one images with resolution of 512×768.

| Model | Peak GPU Memory (GB) | | Training Speed |
| --- | --- | --- | --- |
| | Training | Test | (steps/s) |
| Cheng et al. (2020) | 2.32 | 0.59 | 14.23 |
| Minnen & Singh (2020) | 3.48 | 0.50 | 8.26 |
| Zhu et al. (2022) | 16.84 | 2.10 | 2.66 |
| Zou et al. (2022) | 10.87 | 0.68 | 4.29 |
| Liu et al. (2023) | 14.14 | 1.71 | 2.13 |
| Ours | 12.68 | 1.09 | 3.17 |

### C.4 VISUALIZATION OF CHANNEL ATTENTION WEIGHTS IN T-CA

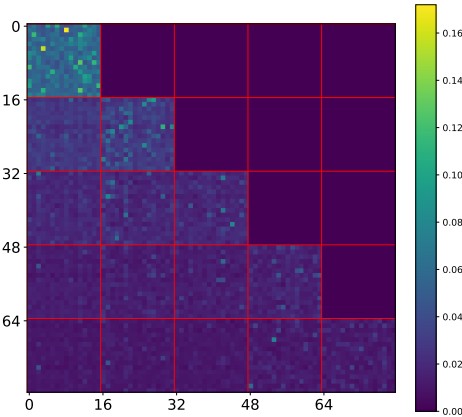

Figure 11: Visualization results of the masked channel attention weights in our T-CA entropy model by inputting *kodim01* in Kodak dataset. We display the average attention weights on all the heads heads in the last transformer layers. Different slices are separated by red lines.

We present a visualization result of the channel attention weights in the last transformer layer of our T-CA entropy model in Figure 11. Each channel attention layers of T-CA consists 16 heads, with each head containing $r * M/n_d = 4 * 320/16 = 80$ channels, resulting in an attention map size of $80 \times 80$. The visualization result explicitly presents the inter-slice and intra-slice dependencies.

#### C.4.1 VISUAL QUALITY RESULTS

Figure 12 and Figure 13 compares the visual quality of the proposed model optimized for MSE. Benefits from the extraction of multi-scale and directional frequency features, our method exhibits superior capability in reconstructing fine details and directional structures. In this way, our models achieve higher compression ratio and better reconstruction quality compared with STF (Zou et al., 2022) and VTM-12.1.

## D LIMITATION AND FUTURE WORK

A potential limitation of our FTIC lies in the fact that our proposed Frequency-Decomposition Window Attention (FDWA) currently supports only two directions (horizontal and vertical). Expanding

the range of supported directions, for example, by incorporating multi-scale and multi-directional analysis like scattering transformation (Patro & Agneeswaran, 2024) could potentially enhance the frequency decomposition of image representation. In addition, our current approach focuses only on image compression, our future work will extend to video compression by exploring the frequency decomposition of spatial-temporal representations (Ding et al., 2022a;b).

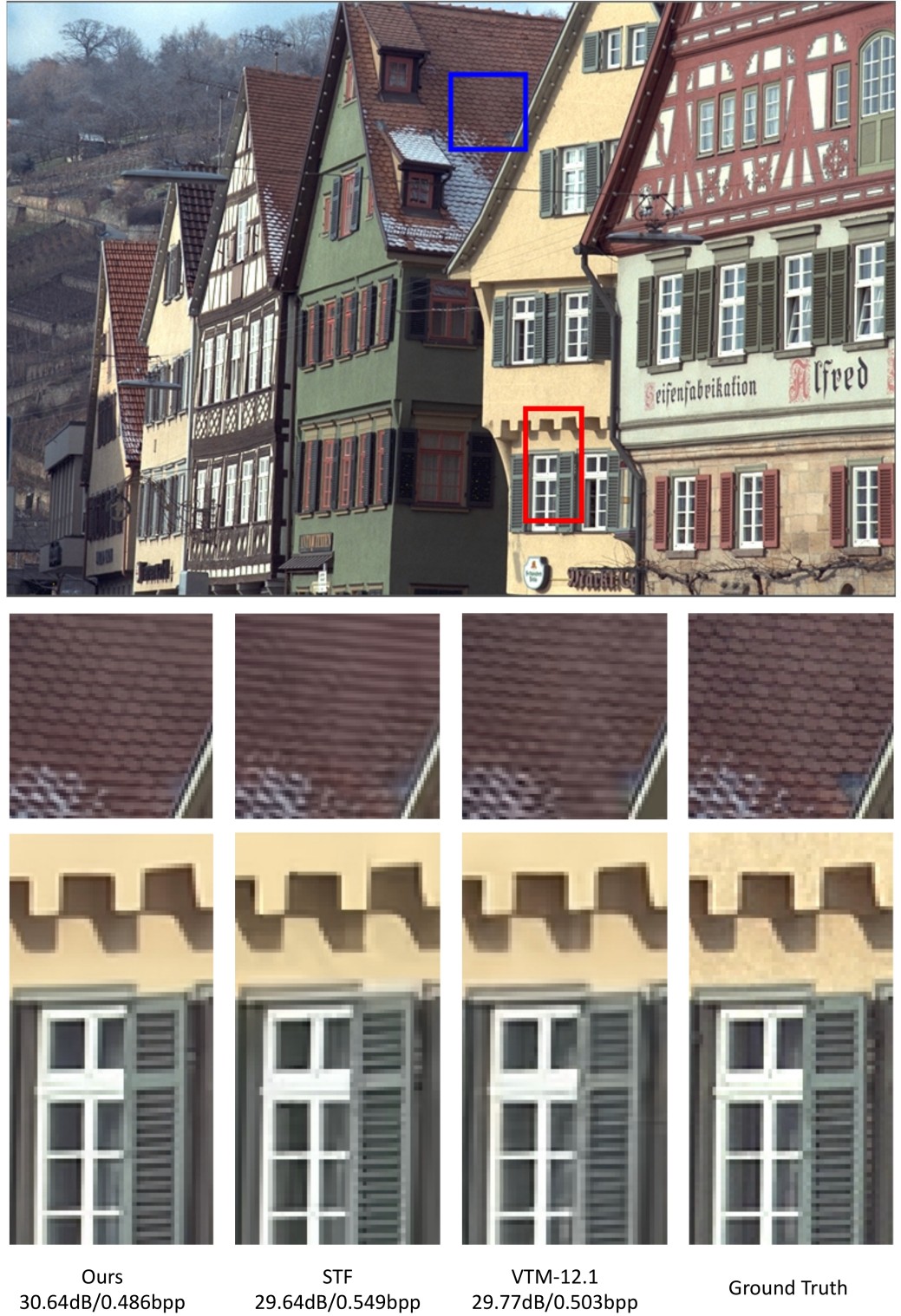

| Ours | STF | VTM-12.1 | Ground Truth |
| 30.64dB/0.486bpp | 29.64dB/0.549bpp | 29.77dB/0.503bpp | |

Figure 12: The visualization of *kodim08* in Kodak dataset by using our FTIC model, STF (Zou et al., 2022) and VTM-12.1. PSNR|bitrate is listed below the subfigures.

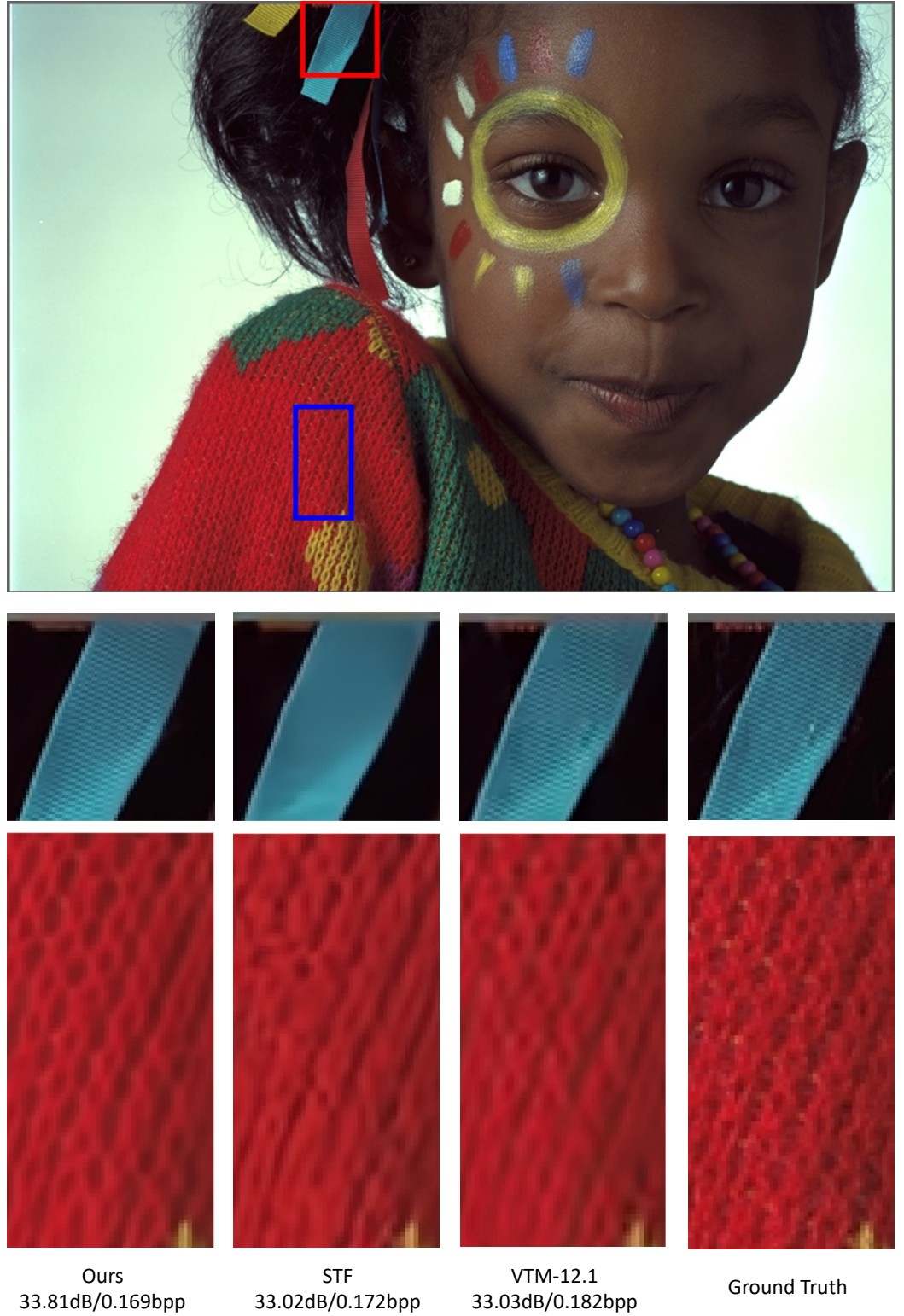

Figure 13: The visualization of *kodim15* in Kodak dataset by using our FTIC model, STF (Zou et al., 2022) and VTM-12.1. PSNR|bitrate is listed below the subfigures.

