# OpenReview forum: "Frequency-Aware Transformer for Learned  Image Compression"
_ICLR.cc/2024/Conference — ICLR 2024 poster_

### Official Review · Reviewer_ZpGq · 2023-10-27

**Soundness:** 3 good
**Presentation:** 3 good
**Contribution:** 3 good
**Rating:** 6
**Confidence:** 5

**Summary:**

The paper studies the frequency problem of learned image compression and develops a frequency-aware method for this problem based on multiscale and directional analysis, called FAT. The method introduces two modules to capture frequency component. Based on FAT, the learned image compression achieves better rate-distortion performance. The method shows improvement over learned codec and conventional codec baselines by a healthy margin.

**Strengths:**

1. The proposed FAT is a novel idea that captures multiscale and directional frequency components and outperforms SOTA.

2. Nice visualization on multiscale and directional decomposition of frequency component. It is an interesting finding that structural information within different frequency ranges also plays a crucial role in learned image compression.

**Weaknesses:**

1. It is interesting to find that FDWA achieves a significant improvement compared to 4x4 blocks and 16x16 blocks. However, it is not clear that how does the multiscale decomposition and the directional decomposition affect. The authors are suggested to provide an ablation study on FDWA.

2. The frequency-aware transformer is realized by two modules, FDWA and FMFFN. Have you tried these two mechanisms in the entropy model? Entropy model plays a crucial role in learned image compression. In my opinion, it is more important to capture frequency component in entropy model. It would be a great improvement if the frequency-aware mechanism works.

**Questions:**

1. Have you ever tried other decomposition ways such as (1) smaller size and bigger size for square window (I am wondering how much improvement can be obtained in this mechanism), or (2) more diverse shapes.

2. What impact does the block size in FMFFN have? When you set the block size responding to the maximum window size in FDWA, does it mean that the FMFFN is used to refine the high-frequence of features?

3. If the decomposition of window size works, how about directly using different sizes of convolution?

---

> ### Author Response · Authors · 2023-11-23
> **Response to Reviewer 2**
>
> We appreciate your valuable comments and insightful  suggestions. We have addressed your concerns as below.
>
> **W1: Ablation Study on FDWA:**
>
> * Thanks for your valuable suggestion. We have performed an ablation study on our FDWA to demonstrate the effect of both multiscale and directional decomposition. Due to the limitations on time and computational resources, we only provide models trained for 500,000 training steps in this response. The Lagrangian multiplier $\lambda$ is set to 0.0483 for all the models.
>
>   In the following table, we use **Multiscale** to denote anisotropic $4\times 4$ and $16\times 16$ windows, and we employ **Directional** to represent isotropic $4\times 16$ and $16\times 4$ windows. Besides, we provide a baseline result with only $4\times 4$ windows for reference. The PSNR, BPP, and R-D loss are averaged across 24 images of the Kodak dataset. Results show that both multiscale and directional decomposition enhance the R-D performance.
>
>
> * | Models            | Multiscale | Directional | PSNR   | BPP    | R-D Loss |
> |-------------------|:------------:|:-------------:|--------|--------|----------|
> | Baseline          | x    | x    | 37.031 | 0.8949 | 1.552    |
> | Multiscale-only   | ✔️         | x  | 37.025 | 0.8722 | 1.537    |
> | Directional-only  | x   | ✔️         | 37.080 | 0.8889 | 1.540    |
> | Proposed          | ✔️         | ✔️          | 37.248 | 0.8910 | 1.517    |
>
>
> **W2: Capturing Frequency Components in Entropy Model:**
>
> * Thanks for your suggestion. We agree that capturing frequency components can further benefit entropy models in LIC, and **we specifically propose a TC-A entropy model to exploit the correlations between the frequency components and reduce the coding budget**, as elaborated in Section~3.3.
>
>   Since the latent representation extracted by the proposed FAT block consists of diverse frequency components across different channels, we employ channel attention in our TC-A entropy model to model the dependencies between these frequency components. Moreover, the channel-wise correlations (*i.e.*, frequencies correlations) captured by our TC-A can adaptively vary with input image, which cannot be achieved by the commonly used CNN-based channel-wise auto-regressive entropy model (*i.e.*, CHARM [R1]).
>
>   **Apply FDWA and FMFFN in Entropy Model:**
>
> * Thanks for your valuable comments. Theoretically, FDWA and FMFFN can be used in the entropy model. Here, we provide additional experimental results based on a recent work [R2]. In this experiment, we adopt the entropy model from [R2] as our baseline and conduct an ablation study to demonstrate the additional benefits of FDWA and FMFFN. The experimental setup follows the ablation study on the TC-A entropy model in Section 4.3 of our paper.
>
>   Despite the Swin-Transformer being only a modest sub-module in the baseline entropy model [R2], our results demonstrate that FDWA and FMFFN can also improve its performance, showing the potential of FDWA and FMFFN in helping achieve more accurate probability distribution estimation.
>   | FDWA         | FMFFN        | BD-rate (Anchor: BPG) |
>   |:--------------:|:--------------:|:------------------------:|
>   |      $\times$ |      $\times$ |         -17.4\%         |
>   |✔️          |      $\times$ |         -18.1\%         |
>   |      $\times$ | ✔️          |         -17.6\%         |
>   |✔️         | ✔️         |         -18.2\%         |
>
>
>   We do not directly apply FDWA and FMFFN in the proposed entropy model (*i.e.*, TC-A) since both these two modules are designed to enhance the Swin-Transformer based on **spatial self-attention**, while our **channel self-attention**-based TC-A is not constructed with the Swin-Transformer.
>
> [R1] Minnen, David, and Saurabh Singh. "Channel-wise autoregressive entropy models for learned image compression." 2020 IEEE International Conference on Image Processing (ICIP). IEEE, 2020.
>
> [R2] Liu, Jinming, Heming Sun, and Jiro Katto. "Learned image compression with mixed transformer-cnn architectures." Proceedings of the IEEE/CVF Conference on Computer Vision and Pattern Recognition. 2023.

---

> ### Author Response · Authors · 2023-11-23
>
> **Q1: Other decomposition ways**
>
> We have tried the decomposition ways mentioned by the reviewer in our earlier experiments, but we did not adopt these configurations in our manuscript due to the out-of-memory (OOM) problem raised by using the bigger size for a square window ($32 \times 32$). Here, we also conduct experiments to comprehensively understand how different decomposition ways affect the performance.
>
> **(1) OOM problem of 32 $\times$ 32 window**: We emphasize that employing a bigger square window ($32 \times 32$) results in significantly increased memory consumption and running time. For example, when we set the window size of **half of the attention heads** as $32\times 32$ and the remaining ones as $2 \times 2$, we meet the OOM problem on our Nvidia RTX 4090Ti GPU (with 24 GB memory) during the training process with each batch containing 8 images with a patch size of 256$\times$ 256.
>
> **(2) Effect of different decomposition ways**: We explored various decomposition ways in this ablation study, including:
>
> - **(a) With Only One Small Window** (*i.e.*, $4\times 4$).
> - **(b) With Multiscale Decomposition**, including two types of windows （*i.e.*,$4\times4$ and $16\times16)$.
> - **(c) With Smaller Size and Bigger Size for Square Window (Reviewer's Suggestion)**, including four types of windows (*i.e.*,$2\times2, 4\times4, 16\times16,$ and $ 32\times32)$.
> - **(d) With Our FDWA**, including four types of window (*i.e.*,$4\times4, 16\times16, 4\times16, 16\times4$).
> - **(e) With More Diverse Shapes (Reviewer's Suggestion)**, including 8 types of window (*i.e.*,$2\times2, 4\times4, 16\times16,  32\times32, 2\times32, 32\times2, 4\times16$, and $16\times4$).
>
> In each of the above methods, the number of attention heads for each type of window attention is identical (*i.e.*, $K/N$ heads, where $K$ denotes the total number of attention heads, and $N$ denotes the number of window types). All the models trained for 500,000 training steps, and the Lagrangian multiplier $\lambda$ for all the models is 0.0483. We observe that the decomposition method employing more diverse shapes achieves slightly lower R-D loss, with a sacrifice in memory consumption and training speed.
>
>
> | **Methods** | **PSNR** | **BPP** | **R-D Loss** | **GPU Memory for Training** | **Training Speed (steps/s)** |
> |:------------:|:--------:|:-------:|:------------:|:---------------------------:|:-----------------------------:|
> |     (a)      |  37.031  |  0.8949 |    1.552     |           10.36GB           |             3.33              |
> |     (b)      |  37.025  |  0.8722 |    1.537     |           13.96GB           |             3.08              |
> |     (c)      |  37.135  |  0.8802 |    1.525     |           20.22GB           |             2.16              |
> |     (d)      |  37.248  |  0.8910 |    1.517     |           12.68GB           |             3.17              |
> |     (e)      |  37.213  |  0.8817 |    1.513     |           15.02GB           |             2.57              |
>
> **Q2: Effect of the Block Size in FMFFN**
>
> The block size in FMFFN influences the balance between modulating local and global features in FMFFN. A block size that is too small in FFT may result in the loss of some global features ( *i.e.,* low-frequency information), while a block size that is too large may impede high-frequency local structure and increase the parameters of the learnable filter matrix.
>
> We want to clarify that FMFFN is not used to refine the high-frequency features only. Setting the block size corresponding to the maximum window size ( *i.e.,* 16$\times$ 16) in FDWA indicates that FMFFN operates within a similar frequency range as FDWA, so that we can comprehensively modulate both low and high-frequency components produced by FDWA. In addition, FFT with 16$\times$ 16 block size also resembles the 16$\times$ 16 block-wise DCT transforms in H.264. Considering these factors, we chose the 16$\times$ 16 block size for FMFFN in our manuscript.
>
> We also conducted ablation studies on the block size of FMFFN, and the results are in the table below. All the models are trained for 500,000 training steps, and the Lagrangian multiplier $\lambda$ for all the models is 0.0483. It indicates that FMFFN with a block size of 8 achieves the best R-D performance in this training set up.
>
> |   Block Size   |   PSNR   |   BPP   |   R-D Loss   |   # Params   |
> |:--------------:|:--------:|:-------:|:-----------------------------:|:------------:|
> |       4        |  37.117  | 0.8812  |            1.524              |   70.44M     |
> |       8        |  37.211  | 0.8789  |            1.511              |   70.55M     |
> |      16        |  37.248  | 0.8910  |            1.517              |   70.97M     |
> |      32        |  37.092  | 0.8793  |            1.529              |   72.55M     |

---

> ### Author Response · Authors · 2023-11-23
>
> **Q3: Convolutions with Diverse Kernel Size.**
> Several works have discussed the impact of convolution kernel sizes in learned image compression. Cheng *et al.* [R3] compare the learned image compression model with diverse convolutional kernel sizes (*i.e.*, $5\times 5$, $7\times 7$, and $9\times 9$) and observe that a larger convolutional kernel size would lead to better R-D performance. Moreover, a larger kernel size could be obtained by stacking small convolutional kernels. Cui *et al.* [R4] employ mask convolutions with diverse kernel sizes { *i.e.*, $3\times 3$, $5\times 5$, and $7\times 7$) to achieve a more precise autoregression model.
>
> However, we have not found discussions on anisotropic kernel size or analyses on the effect of kernel size from the perspective of frequency decomposition for convolution-based LIC models. Some works [R5, R6, R7] show that convolutions with a local receptive size tend to present the characteristics of a high-frequency filter. While there is potential to achieve frequency decomposition with diverse kernel sizes of convolutions, achieving much larger kernel sizes may pose challenges in computation complexity or require stacking too many small kernel convolution layers.
>
> We highlight that our FDWA can achieve multiscale and directional frequency decomposition **in each attention layer in a more effective and simple way.**
>
> - [R3] Z. Cheng, H. Sun, M. Takeuchi, and J. Katto, "Deep Residual Learning for Image Compression," CVPR Workshops. 2019.
>
> - [R4] Z. Cui, J. Wang, S. Gao, T. Guo, Y. Feng, and B. Bai, "Asymmetric Gained Deep Image Compression With Continuous Rate Adaptation," 2021 IEEE/CVF Conference on Computer Vision and Pattern Recognition (CVPR), Nashville, TN, USA, 2021, pp. 10527-10536.
>
> - [R5] Park, Namuk, and Songkuk Kim. "How Do Vision Transformers Work?." International Conference on Learning Representations. 2022.
>
> - [R6] Yin, Dong, et al. "A fourier perspective on model robustness in computer vision." Advances in Neural Information Processing Systems 32 (2019).
>
> - [R7] Wang, Haohan, et al. "High-frequency component helps explain the generalization of convolutional neural networks." Proceedings of the IEEE/CVF conference on computer vision and pattern recognition. 2020.
>
> We sincerely appreciate your dedicated review of our paper. The author response period for this submission has been extended until December 1st, and we are looking forward to your further response.

---

### Official Review · Reviewer_fcgY · 2023-10-31

**Soundness:** 3 good
**Presentation:** 3 good
**Contribution:** 2 fair
**Rating:** 6
**Confidence:** 4

**Summary:**

To overcome the problem of existing LIC (Learned Image Compression) methods are redundant in latent representation, this paper suggests a nonlinear transformation and makes the following three improvements:

1)	This paper proposes a frequency-decomposition window attention (FDWA), which leverages diverse window shapes to capture frequency components of natural images.

2)	This paper develops a frequency-modulation feed-forward network (FMFFN) that adaptively ensembles frequency components for improved R-D performance.

3)	This paper presents a transformer-based channel-wise autoregressive model (T-CA) for effectively modeling dependencies across frequency components.

Experimental results show that this paper achieves state-of-the-art R-D performance on several datasets.

**Strengths:**

1) The paper is overall easy to understand and clearly written. One of the primary strengths of this paper is the claimed SOTA rate-distortion performance.
2) The authors found that existing learned image compression methods lead to potential representation redundancy due to limitations in capturing the anisotropic frequency components of anisotropy and preserving directional details. Some attempts including FMFFN and FDWA are proposed to address this issue.

**Weaknesses:**

1) The idea of applying frequency processing to the learned compression framework is not new. For example, conv in the frequency domain [1] has been used in Balle’s early work [2]. Wavelet-based compression framework has also been proposed [3]. The authors should cite, compare, and identify their differences.

[1] Rippel, Oren, Jasper Snoek, and Ryan P. Adams. "Spectral representations for convolutional neural networks." Advances in neural information processing systems 28 (2015).

[2] Ballé, Johannes, Valero Laparra, and Eero P. Simoncelli. "End-to-end optimized image compression." arXiv preprint arXiv:1611.01704 (2016).

[3] Ma, Haichuan, et al. "iWave: CNN-based wavelet-like transform for image compression." IEEE Transactions on Multimedia 22.7 (2019): 1667-1679.

2) The authors should compare with more existing works (e.g. [3,4]) to demonstrate the SOTA performance of the paper.

[4] Liu J, Sun H, Katto J. Learned image compression with mixed transformer-cnn architectures[C]//Proceedings of the IEEE/CVF Conference on Computer Vision and Pattern Recognition. 2023: 14388-14397.

[5] Fu H, Liang F, Liang J, et al. Asymmetric Learned Image Compression with Multi-Scale Residual Block, Importance Scaling, and Post-Quantization Filtering[J]. IEEE Transactions on Circuits and Systems for Video Technology, 2023.

**Questions:**

1. Table 1 and Table 5 in the article involve flop calculations, may I ask the authors what methods or tools they used to calculate the complexity? As far as I know, many existing tools that count complexity do not calculate correctly the complexity of the internal operators of the transformer. Besides, Table 5 is kind of confusing. Can the authors explain what the (1), (2), (3) in the table mean here.

2. Adding a transformer on the codec side would result in a longer training time and a larger memory requirement that is not hardware-friendly for the actual deployment of the compression model. Can the authors provide the training time as well as the corresponding average and peak memory consumption for both training and test? Besides, the three methods compared in Table 5 are all transformer-based methods, and there is no comparison of decoding time with, for example, Minnen[1] and GMM[2], where both the encoder and decoder use a CNN structure.

[1] Minnen D, Singh S. Channel-wise autoregressive entropy models for learned image compression[C]//2020 IEEE International Conference on Image Processing (ICIP). IEEE, 2020: 3339-3343.

[2] Cheng Z, Sun H, Takeuchi M, et al. Learned image compression with discretized gaussian mixture likelihoods and attention modules[C]//Proceedings of the IEEE/CVF conference on computer vision and pattern recognition. 2020: 7939-7948.

---

> ### Author Response · Authors · 2023-11-15
> **Response to Reviewer fcgY**
>
> We appreciate your insightful comments and valuable suggestions. We address your concerns as below.
>
> **W1: Novelty of this paper.**
>
> - First of all, we would like to clarify that **the novelty of this paper is the novel method to exploit multi-scale and directional frequency information in nonlinear transforms rather than employing frequency-based operations.** We HAVE NOT claimed that our work is the first to apply frequency-based operations in LIC. On the contrary, we break the limitation of existing transformer-based LIC models that cannot achieve multi-scale and directional analysis due to the isotropic window attention.
>
> - Subsequently, we have discussed the three papers [R1, R2, R3] mentioned by the reviewer and clarified the difference between this paper and [R1, R2, R3]. In summary, [R1] and [R2] **aim to accelerate network optimization** and directly learn the parameters of CNN in the spectral domain. [R3] substitutes the filters in the traditional separable wavelet transforms with trained CNNs to improve the R-D performance of  JPEG2000. **These methods cannot achieve multi-scale directional analysis by leveraging isotropic DFT [R1] and DCT [R2] bases or introducing CNNs into 1-D Wavelet transforms [R3].** By contrast, in this paper, we propose to capture diverse frequency components with the frequency-aware transformer and achieve an end-to-end optimized multi-scale and directional analysis by anisotropic window attention. We elaborate our differences from [R1, R2, R3] as below.
>
>   * **Difference from [R1] and [R2].** [R1] and [R2] parameterize CNNs in the frequency domain for fast convergence since spatial domain convolutions are equal to spectral domain multiplications. DCT [R1] and DFT [R2] are adopted as the transforms for the linear filters in CNNs. However, such parameterization only affects convergence speed in optimizing the parameters but does not change the network architectures.  It should be noted that this idea is also considered for speeding up various tasks such as  image classification [R1], inverting deep networks [R6], and mobile sensing [R7], rather than specifically designed for extracting frequency components in LIC.
>
>     In contrast, our approach captures and processes diverse frequency components to achieve more  compact image representations for improved R-D performance.  **Different from [R1] and [R2], we design FMFFN specifically for LIC to to adaptively decide the needed frequency components** and further eliminate redundancy across different components, though FFT is used to transform the features into frequency domain.**
>
>   * **Difference from [R3].** iWave [R3] is developed based on the JPEG2000 schemes by substituting the discrete wavelet transforms (*e.g.*, Daubechies 5/3 and 9/7 wavelets) with CNNs. iWave [R3] cannot be end-to-end optimized by fixing the entropy model (*i.e.*, the probability distribution of discretized symbols) for entropy coding and yields limited R-D performance gain over traditional JPEG2000.  iWave++ [R8] is later developed to enable end-to-end learning for LIC. However,  iWave [R3] and iWave++ [R8] realize  nonlinear transform by stacking1-D transforms on horizontal and vertical directions, inspired by JPEG2000 that leverages 2-D separable wavelet transforms realized independently on the horizontal and vertical directions. As a result, they cannot realize multi-dimensional directional transform and is restricted in achieving compact representation for image compression
>   * **Different from [R3] and [R8] that independently perform 1-D transforms, our method achieves directional transforms that process the horizontal and vertical directions simultaneously.**  We are the first to leverage the transformer to exploit the different frequency components, which has not been achieved in existing LIC model. Furthermore, we  propose the T-CA entropy model to model the correlations across different frequency components for end-to-end optimization.
>
> We have cited these works as. Ball'e et al. (2016), Rippel et al. (2015), and Ma et al. (2019; 2020) and discussed them in the related work section in our revision.
>
> **W2: SOTA performance.**
> * We exactly achieve the SOTA performance and outperform [R4] and [R5] mentioned by the reviewer. [R4] is the current SOTA method for LIC and has been compared in the previously submitted manuscript. We show that, compared with [R4], we yield 2.6% BD-rate saving on the Kodak dataset. Moreover, [R5] is inferior to [R4] in R-D performance. Compared with VVC, [R5] yields 2.1% BD-rate reduction on the Kodak dataset, while **we can achieve a significantly higher 14.5% BD-rate reduction**. We have included the results for [R5] in Fig.~4 in the revised manuscript.

---

> ### Author Response · Authors · 2023-11-15
>
> **Q1: We address the concerns on Tables 1 and 5 as below.**
>
> **Table 5 for calculating the complexity:**
> Thanks for your kind remainder. We noticed that *deepspeed* used in the previous transformer-based LIC work [R9] ignores the matrix multiplication and softmax operation of self-attention. Thus, in our evaluation, we used  the *FlopCountAnalys* function from *fvcore* library, which can appropriately process these internal operations of the transformer.
>
> **Presentation of Table 1:**
> We have revised Table 1 to improve its clarity. Here, (1) (2) (3) mean three different settings of architectures of FAT Block, and the number only represents the indices of different architectures.
>
> **Q2: Comparison with CNN-based methods**
>
> **The integration of transformers into LIC is a notable trend for enhancing compression performance.** However, existing transformer-based LIC methods cannot achieve multiscale and directional analysis, which is an important and indispensable topic spanning from conventional codecs to recent LIC methods. Therefore, in this paper, we propose a frequency-aware transformer for LIC to further enable multiscale and directional analysis. To this end, we focus on **achieving better R-D performance with equivalent complexity over existing transformer-based LIC.**
>
> According to the reviewer's comments, we compare our method with existing CNN-based [R10, R11] and transformer-based methods [R4, R9, R12] in terms of training speed, memory requirement, and FLOPs in Tables 5 and 6 in the revision (also seen as below). Note that all the existing methods claim that they require 2M$\sim$3.5M training steps in total for convergence. Compared with CNN-based methods, transformer-based methods exhibit evidently high R-D performance at the cost of memory consumption, training speed, and coding time. Notably, compared with existing transformer-based methods, the proposed method achieves apparent R-D performance improvement with only a slightly higher computational complexity.  In addition, many existing works [R13,R14,R15] focus on achieving efficient transformer, these facts imply the potential of our method in further exploring the efficient transformer-based LIC.
>
>
>
> | **Model**      | **GFLOPs (Enc.)** | **GFLOPs (Dec.)** | **Inference Latency (Enc.) (ms)** | **Inference Latency (Dec.) (ms)** | **#Params** | **BD-rate** |
> | ----------------------------- | :------: |:-:| :-: |:-:|:-:|-:|
> | Cheng et al. (2020) [R10]    | 154               | 229               | $>$1000                          | $>$1000                          | 26.60M      | 3.6%        |
> | Minnen & Singh (2020)  [R11]  | 101               | 100          | 56                               | 43                               | 55.13M      | 1.1%        |
> | Zhu et al. (2022) [R9] | 116              | 116               | 110                              | 99                               | 32.71M      | -3.3%       |
> |   Zou et al. (2022) [R12]       | 285               | 276               | 97                               | 101                              | 99.58M      | -4.3%       |
> | Liu et al. (2023)  [R4]     | 317               | 453               | 255                              | 322                              | 76.57M      | -11.9%      |
> | Ours                          | 141               | 349               | 125                              | 242                              | 70.97M      | -14.5%      |
> | VTM-12.1                       | -                 | -                 | -                                | -                                | -           | 0%          |
>
> | **Model**                       | **Peak GPU Memory (GB) for Training** | **Peak GPU Memory (GB) for Test** | **Training Speed (steps/s)** |
> | :------------------------------- | :-----------------------------------:| :--------------------------------: | :-----------------------------: |
> | Cheng et al. (2020) [R10]       | 2.32                                | 0.59                             | 14.23                         |
> |Minnen & Singh (2020)  [R11]     | 3.48                                | 0.50                             | 8.26                          |
> | Zhu et al. (2022) [R9] | 16.84                               | 2.10                             | 2.66                          |
> |   Zou et al. (2022) [R12]           | 10.87                               | 0.68                             | 4.29                          |
> | Liu et al. (2023)  [R4]          | 14.14                               | 1.71                             | 2.13                          |
> | Ours                            | 12.68                               | 1.09                             | 3.17                          |
>
>
> We sincerely thank you for your efforts in reviewing this paper, and we are looking forward to your response.

---

> ### Author Response · Authors · 2023-11-17
> **Reference**
>
> **Reference**
> - [R1] Rippel, Oren, Jasper Snoek, and Ryan P. Adams. "Spectral representations for convolutional neural networks." Advances in neural information processing systems 28 (2015).
> - [R2] Ballé, Johannes, Valero Laparra, and Eero P. Simoncelli. "End-to-end optimized image compression." arXiv preprint arXiv:1611.01704 (2016).
> - [R3] Ma, Haichuan, et al. "iWave: CNN-based wavelet-like transform for image compression." IEEE Transactions on Multimedia 22.7 (2019): 1667-1679.
> - [R4] Liu, Jinming, Heming Sun, and Jiro Katto. "Learned image compression with mixed transformer-cnn architectures." Proceedings of the IEEE/CVF Conference on Computer Vision and Pattern Recognition. 2023.
> - [R5] Fu, Haisheng, et al. "Asymmetric Learned Image Compression with Multi-Scale Residual Block, Importance Scaling, and Post-Quantization Filtering." IEEE Transactions on Circuits and Systems for Video Technology (2023).
> - [R6] Wong, Eric, and J. Zico Kolter. "Neural network inversion beyond gradient descent." Advances in Neural Information Processing Systems, Workshop on Optimization for Machine Learning. 2017.
> - [R7] Yao, Shuochao, et al. "Deepsense: A unified deep learning framework for time-series mobile sensing data processing." Proceedings of the 26th international conference on world wide web (WWW). 2017.
> - [R8] Ma, Haichuan, et al. "End-to-end optimized versatile image compression with wavelet-like transform." IEEE Transactions on Pattern Analysis and Machine Intelligence 44.3 (2020): 1247-1263.
> - [R9] Zhu, Yinhao, Yang Yang, and Taco Cohen. "Transformer-based transform coding." International Conference on Learning Representations. 2022.
> - [R10] Cheng, Zhengxue, et al. "Learned image compression with discretized gaussian mixture likelihoods and attention modules." Proceedings of the IEEE/CVF conference on computer vision and pattern recognition. 2020.
> - [R11] Minnen, David, and Saurabh Singh. "Channel-wise autoregressive entropy models for learned image compression." 2020 IEEE International Conference on Image Processing (ICIP). IEEE, 2020.
> - [R12]Zou, Renjie, Chunfeng Song, and Zhaoxiang Zhang. "The devil is in the details: Window-based attention for image compression." Proceedings of the IEEE/CVF conference on computer vision and pattern recognition. 2022.
> - [R13] Zhang, Xindong, et al. "Efficient long-range attention network for image super-resolution." European Conference on Computer Vision. Cham: Springer Nature Switzerland, 2022.
> - [R14] Pan, Zizheng, Jianfei Cai, and Bohan Zhuang. "Fast vision transformers with hilo attention." Advances in Neural Information Processing Systems 35 (2022): 14541-14554.
> - [R15] Li, Yanyu, et al. "Rethinking vision transformers for mobilenet size and speed." Proceedings of the IEEE/CVF International Conference on Computer Vision. 2023.

---

> > ### Comment · Reviewer_fcgY · 2023-11-22
> > **Response to the authors' comments**
> >
> > Thank you for your response. The overall evaluation, with these supplemented results and explanations, is comprehensive. Despite my reservations regarding the novelty and contribution of this paper, I have decided to raise my rating.

---

> > > ### Author Response · Authors · 2023-11-23
> > >
> > > We are grateful for your comments and suggestions in enhancing the quality of our paper and for your decision to update the score. We will try our best to further illustrate the novelty of our paper later.

---

### Official Review · Reviewer_kfZs · 2023-11-22

**Soundness:** 3 good
**Presentation:** 3 good
**Contribution:** 3 good
**Rating:** 6
**Confidence:** 4

**Summary:**

This paper aims to reduce redundancies in the latent representation of Learned Image Compression methods.
For this purpose it proposes a new frequency-aware transformer block which utilizes two new components.
1. FDWA: a window attention block to capture various frequency components
2. FMFFN: a block which modulates the frequency components

Additionally this paper proposes a transformer-based channel-wise autoregressive entropy model (T-CA)
In combination these methods achieve SOTA performance on various datasets.

**Strengths:**

1. The method achieves SOTA performance without unreasonable performance cost.
2. The paper is well written and provides nice visualizations of core ideas.

**Weaknesses:**

There are no explicit ablations of some design decisions of the FAT block. What effect do the relative window sizes of the FDWA module have? What's the impact of omitting some of the windows (eg. omitting vertical/horizontal windows)?

There are typos in the caption of Figure 3 (p.6) and Table 2 (p.8)

**Questions:**

How does the use of the T-CA entropy model affect inference times (compared to existing models such as CHARM)?

---

> ### Author Response · Authors · 2023-11-27
> **Response to Reviewer kfZs**
>
> **W1:Effect of the Relative Window Sizes of the FDWA Module:**
>
> Thanks for your valuable suggestions. We have performed an ablation study on our FDWA to demonstrate the effect on the relative window sizes. Due to the time limit, we only provide models trained for 500,000 training steps in this response. The Lagrangian multiplier $\lambda$ is set to 0.0483 for all the models.
>
> - **Anisotropic and Isotropic Windows.**
>
>   In the following table, we use **Multiscale** to denote isotropic $4\times 4$ and $16\times 16$ windows, which aim at extracting multiscale frequency information. Besides, we employ **Directional** to represent anisotropic $4\times 16$ and $16\times 4$ windows, which are designed to capture the crucial directional frequency information. We also provide a baseline result with only $4\times 4$ windows for reference.
>   The PSNR, BPP, and R-D loss are averaged across 24 images of the Kodak dataset. Results show that both isotropic and anisotropic windows enhance the R-D performance.
> | Models            | Multiscale | Directional | PSNR   | BPP    | R-D Loss |
> |-------------------|:------------:|:-------------:|--------|--------|----------|
> | Baseline          | x    | x    | 37.031 | 0.8949 | 1.552    |
> | Multiscale-only   | ✔️         | x  | 37.025 | 0.8722 | 1.537    |
> | Directional-only  | x   | ✔️         | 37.080 | 0.8889 | 1.540    |
> | Proposed          | ✔️         | ✔️          | 37.248 | 0.8910 | 1.517    |
>
> - **Anisotropic (Directional) Windows**
>
>   We have also conducted an ablation study on the vertical and horizontal windows. The following table shows the R-D performance of the FDWA-based models without horizontal or vertical windows. For the model without horizontal windows (*i.e.*, w/o horizontal windows), we replace the original horizontal windows ($4\times 16$) in the FDWA with vertical windows ($16\times 4$). The case is vice versa for the model without vertical windows (*i.e.*, w/o vertical windows). Results show that removing either horizontal or vertical windows leads to performance degradation, and incorporating these two anisotropic windows leads to the best R-D performance.
>
> |   Models                  | PSNR   | BPP    | R-D Loss |
> |-------------------------|--------|--------|----------|
> | Proposed FDWA           | 37.248 | 0.8910 | 1.517    |
> | w/o horizontal windows | 37.134 | 0.8822 | 1.524    |
> | w/o vertical windows   | 37.158 | 0.8874 | 1.526    |
>
> **W2:Typos.**
>
>    Thanks for your kind reminder. In our revised manuscript, we have corrected the typos in the caption of Figure 3 (p.6) and Table 2 (p.8).

---

> ### Author Response · Authors · 2023-11-28
> **Response to Reviewer kfZs （Part2)**
>
> **Q: Inference Times with Different Entropy Models**
>
> The following table compares the inference latency during the encoding and decoding process of our framework by using several entropy models, including CHARM [R1], CHARM with SWAtten (*i.e.,* the entropy model of [R2]), and our T-CA entropy model. The number of slices is set to 5 for all models to guarantee a fair comparison. Note that the inference time of our T-CA in this response is slightly different from the results reported in our previously submitted paper, as we perform this evaluation on a new workstation with CPUs and GPUs identical to the previous one.
>
> The experimental results indicate that our T-CA has a comparable inference time in encoding and less than 2x in decoding compared to CHARM [R1]. In contrast, CHARM with SWAtten [R2] exhibits an inference time of more than 2x for encoding and decoding compared to CHARM.
>
>
> | Entropy Model         | Inference Latency (Enc.) (ms) |Inference Latency (Dec.) (ms)
> |-----------------------|:------------------------:|:-:|
> | CHARM [R1]            | 92                     | 97                     |
> | CHARM + SWAtten [R2]  | 196                    | 206                    |
> | Our T-CA              | 103                    | 189                    |
>
>
> We want to address the fact that the proposed T-CA entropy model achieves the best R-D performance among all three entropy models, although it is not the most complicated entropy model.
>
>
> Compared to BPG, our T-CA achieves a significant **19.2%** BD-rate reduction on the Kodak dataset, while CHARM and CHARM with SWAtten only yield **14.2%** and **17.4%** BD-rate reduction, respectively. The experimental setup follows the ablation study on the TC-A entropy model in Section 4.3 of our paper.
>
> The experiments on inference latency and R-D performance show that our T-CA can estimate the probability distribution more precisely with a reasonable increase in computational complexity.
>
> - [R1] Minnen, David, and Saurabh Singh. "Channel-wise autoregressive entropy models for learned image compression." 2020 IEEE International Conference on Image Processing (ICIP). IEEE, 2020.
>
> - [R2] Liu J, Sun H, Katto J. Learned image compression with mixed transformer-cnn architectures. Proceedings of the IEEE/CVF Conference on Computer Vision and Pattern Recognition. 2023: 14388-14397.

---

### Official Review · Reviewer_Xwct · 2023-11-26

**Soundness:** 3 good
**Presentation:** 3 good
**Contribution:** 2 fair
**Rating:** 6
**Confidence:** 3

**Summary:**

A learned image compression approach based on transformer architecture is presented. The main innovation is the introduction of a frequency-aware module that performs multi-scale, directional analysis of the transformer features, and then uses an FFT to weight the important frequency components. Results show mild to moderate improvements in the RD-curve and BD-metric on various datasets.

**Strengths:**

The intuition behind the method is sound. Writing and explanation is clear. Results show improvements over various transformer-based baselines.

**Weaknesses:**

This is a general comment rather than a specific weakness. It is not entirely clear how much improvement is brought about by the specific structure of the FDWA and FMFFN blocks, rather than the fact that some learnable layers have been included which would increase the overall capacity of the network. The authors have provided a few ablations in Figures 5. and 6. which do somewhat support their claim, but I feel this could be made stronger in future versions. For instance,

1. What if additional windows with different aspect ratios were used? Would these provide further improvement? This could be achieved by either increasing $K$ (the number of heads), or by using the same number of heads but partitioning into more groups to accommodate the different aspect ratios?

2. The impact of the FMFFN block seems to be rather small. What if instead of a block FFT, a learnable conv layer (and deconv for the IFFT) was used?

**Questions:**

Included in comment above.

---

> ### Author Response · Authors · 2023-12-01
> **Response to Reviewer Xwct  (Part 1)**
>
> Thanks for your valuable suggestions. We further clarify the improvement brought by the proposed modules, and provide additional experiments on the window shapes.
>
> **Q1: Improvement brought by FDWA and FMFFN**
>
> We want to clarify that our FDWA and FMFFN only **introduce less than 1% increase in model parameters** compared with the baseline, and the performance improvement does come from their specific and well-designed structure. In Table 1 of the revised manuscript, we compare the computational complexity, number of parameters, and R-D performance of three transformer-based transforms to show the superiority of the proposed FDWA and FMFFN. The four included transformer-based transforms are as follows.
>
> - SwinT ($4\times4$): Swin-Transformer with $4\times4$ windows.
> - SwinT ($16\times16$): Swin-Transformer with $16\times16$ window.
> - FAT w/o FMFFN: Swin-Transformer with FAT blocks of FDWA only.
> - FAT: Swin-Transformer with full FAT blocks, including FDWA and FMFFN.
>
>
> | Methods             | FDWA | FMFFN | **GFLOPs** | **#Params** | **BD-rate** |
> |---------------------|------|-------|------------|-------------|-------------|
> | SwinT ($4\times4$)  |    x   |     x   | 77.3       | 70.29M      | -9.3%       |
> | SwinT ($16\times16$)|  x  |    x    | 91.3       | 70.50M      | -11.2%      |
> | FAT w/o FMFFN       | ✔️  |    x    | 82.2       | 70.36M      | -13.3%      |
> | FAT                 | ✔️  | ✔️     | 83.1       | 70.97M      | -14.5%      |
> | VTM-12.1            | -    | -     | -          | -           | 0%          |
>
> The above table shows that the proposed FDWA and FAT achieve significant R-D performance improvement (at most -5.2\% BD-rate reduction) with negligible increase in the number of parameters (within 1\%). Therefore, the increased model capacity is not the main reason for the evident performance enhancement. Moreover, both FDWA and FMFFN are indispensable components of the proposed FAT, contributing to the R-D performance enhancement.
>
> **Q2: Additional Windows with Different Aspect Ratios**
>
> Firstly, we want to clarify that we employed the same total number of attention heads and partitioned them into more groups to accommodate various window types in ablation studies.
>
> Moreover, we have finished experiments on FAT with diverse window shapes. The four groups of attention windows used in the experiments are as follows, where the total number of attention heads is identical.
>
> - (a) Window attention with $4\times 4$ windows.
> - (b) Window attention with the proposed FDWA that comprises of four types of windows (*i.e.*, $4\times4, 16\times16, 4\times16, 16\times4$).
> - (c) Window attention with more diverse types of windows (*i.e.*, $ 4\times4, 16\times16,  4\times16, 16\times4, 4\times16$, $8\times4$, $8\times16$, and $16\times8$).
> - (d) Window attention with larger aspect ratios (*i.e.*, $2\times2, 4\times4, 16\times16,  32\times32, 2\times32, 32\times2, 4\times16$, and $16\times4$).
>
> | **Methods** | **PSNR** | **BPP** | **R-D Loss ↓** | **GPU Memory for Training** | **Training Speed (steps/s)** |
> |:-------------|:----------:|:---------:|:--------------:|:-----------------------------:|:-------------------------------:|
> | (a)         | 37.031   | 0.8949  | 1.552        | 10.36GB                     | 3.33                          |
> | (b)         | 37.248   | 0.8910  | 1.517        | 12.68GB                     | 3.17                          |
> | (c)         | 37.176   | 0.8838  | 1.520        | 13.95GB                     | 2.84                          |
> | (d)         | 37.213   | 0.8817  | 1.513        | 15.02GB                     | 2.57                          |
>
> All the models were trained for 500,000 training steps due to the time limit, and the Lagrangian multiplier $\lambda$ for all the models is 0.0483. The PSNR, BPP, and R-D loss are averaged across 24 images of the Kodak dataset. The comparison results are presented in the above table.  We observe that employing more diverse window shapes (*i.e.*, method (c)) does not bring further improvement compared with our FDWA. Meanwhile, method
> (d) employing windows with larger aspect ratios achieves slightly lower R-D loss, with a sacrifice in memory consumption and training speed. Besides, the proposed FDWA can achieve **a better trade-off between R-D performance and overall complexity**.

---

> ### Author Response · Authors · 2023-12-01
> **Response to Reviewer Xwct (Part 2)**
>
> **Q3: Impact of FMFFN**
>
> The FMFFN is a **simple yet effective module** of the proposed framework, which adaptively modulates the frequency components, further bringing a 1.2% BD-rate reduction with negligible complexity overhead.
>
> The quantitative results in Fig. 6 of the manuscript also illustrate the effect of FMFFN. FMFFN can help the high-bit-rate model reserve more high-frequency information while guiding the low-bit-rate model to focus more on the low-frequency information. Thus, the redundancies in the latent representation can be further reduced.
>
> **Replace FFT with Convolutional Layers**
>
> Thanks for your suggestion. First, we clarify the necessity of the block-wise FFT/IFFT in FMFFN:
>
> 1. **FFT and IFFT** constitute a reversible transformation pair, guaranteeing that frequency modulation with matrix $W$ in the frequency domain is a comprehensive and meaningful operation.
>
> 2. **Block operation** further supports applying frequency modulation operation to input images with different resolutions, as the size of block-wise filter $W$ only depends on the block size.
>
> To further demonstrate the effect of FMFFN, we have conducted experiments to replace the non-trainable block-wise FFT/IFFT with trainable convolution/deconvolution (Conv/Deconv) layers. We also drop our filter $W$ for trainable Conv/Deconv layers since the linear weight filter has been integrated into the Conv layer.
>
> However, introducing Conv and Deconv layers **with stride** in the feedforward network (FFN) led to instability in the model convergence.
> We used two $3\times 3$ Conv layers **without stride** as a substitute for FFT/IFFT. We have also conducted another experiment using two $3\times 3$ depth-wise Conv layers to prevent a significant parameter increase. The results are shown in the Table below.
>
> | **Methods**                        | **PSNR** | **BPP** | **R-D Loss ↓** | **#Params** |
> |-------------------------------------|----------|---------|--------------|-------------|
> | block FFT/IFFT with $W$             | 37.248   | 0.8910  | 1.517        | 70.97M      |
> | two Conv layers                     | 36.993   | 0.8624  | 1.532        | 84.79M      |
> | two depth-wise Conv layers          | 37.003   | 0.8641  | 1.534        | 70.52M      |
>
>
> All the models were trained for 500,000 training steps due to the time limit, and the Lagrangian multiplier $\lambda$ for all the models is 0.0483. From the above table, we find the following conclusions.
>
> - Replacing the block-wise frequency modulation operation with the trainable Conv layers leads to a **degradation** in R-D performance, showing the crucial effect of our FMFFN.
>
> - Increasing the overall capacity of the model may **not** bring much performance improvement, suggesting that the performance improvement of FMFFN and FDWA comes from their specific structure.

---

### Author Response · Authors · 2023-11-15
**Modifications to the revision**

We sincerely appreciate your efforts on review works. All the revised contents in the revision are colored in red and the main changes are summarized as follows:


**Related Works:**

1.We incorporate references [1][2][3][4] into our related works and discuss they in this section.



**Experiments:**

1.We include the RD-curve of [5] as Fu(TCSVT2023) in the Fig.4 to showcase our SOTA performance.

2.We modify the structure of Tab.1 to clearly illustrate the different variants in our ablation study.

3.We modify the values of Peak GPU Memory in Tab.1, because our previous statistics incorrectly enabled gradients (*requires\_grad=True*), leading to higher GPU memory usage during the evaluation stage.



**Appendix:**

1.We add the comparison to other CNN-based LIC methods in Tab.5.

2.We add Tab.6 to compare the training speed and GPU memory requirements of our method with existing CNN-based and transformer-based LIC methods.

[1] Ballé J, Laparra V, Simoncelli E P. "End-to-end Optimized Image Compression." International Conference on Learning Representations. 2016.

[2] Rippel, Oren, Jasper Snoek, and Ryan P. Adams. "Spectral representations for convolutional neural networks." Advances in neural information processing systems 28 (2015).


[3] Ma, Haichuan, et al. "iWave: CNN-based wavelet-like transform for image compression." IEEE Transactions on Multimedia 22.7 (2019): 1667-1679.

[4] Ma H, Liu D, Yan N, et al. "End-to-end optimized versatile image compression with wavelet-like transform". IEEE Transactions on Pattern Analysis and Machine Intelligence, 2020, 44(3): 1247-1263.

[5] Fu H, Liang F, Liang J, et al. "Asymmetric Learned Image Compression with Multi-Scale Residual Block, Importance Scaling, and Post-Quantization Filtering". IEEE Transactions on Circuits and Systems for Video Technology, 2023.

---

### Author Response · Authors · 2023-11-30
**Modifications to the revision (part2)**

We sincerely appreciate your efforts on review works.  We update the our revision and  the revised contents re colored in red. The main changes are summarized as follows:

**Experiments:**

We provide the model parameter count of each variant of Table 1.

**Typos:**

We correct the typos in the caption of Figure 3 (p.6) and Table 2 (p.8).

---

### Author Response · Authors · 2023-12-02
**General Comment**

We would like to express our gratitude to all the reviewers for their thoughtful reviews; we truly appreciate the time they took to carefully analyze the paper.

**Novelty of our paper**

Here, we further summarize the novelty and contributions of our paper.
Frequency decomposition is a hot topic in both traditional image compression and current learned image compression (LIC), as discussed in the section of related work in the paper. Difference of our work from existing methods is highlighted as below.

1. Recent state-of-the-art (SOTA) works [R1, R2, R3] in LIC tend to use more expressive transforms (e.g., transformer-based architecture) to achieve better R-D performance but are limited in faithfully recovering the original image from its compact representation (*i.e.*, reconstruction fidelity). Compared to these SOTA works, **we  are the first to achieve multiscale and directional frequency decomposition in the transformer to further reduce the redundancies in the latent representation without increasing complexity.** Additionally, we propose an FMFFN that adaptively modulates frequency components and introduce a novel TC-A entropy model that explores cross-frequency correlations to achieve SOTA R-D performance.

2. Compared with the lifting scheme-based methods iWave [R4, R5] mentioned by reviewer fcgY, our FAT-LIC has the following differences and advantages:

   - Our FDWA can achieve directional transforms that process the horizontal and vertical directions **simultaneously**. In contrast, [R4, R5] must **alternately and independently** perform **1-D transforms** in the horizontal and vertical directions, thus retaining directional redundancies.

   - Our method leverages varying scales for the horizontal and vertical directions by the resizable rectangular window, which can be viewed as neural network based extension of multi-scale geometry analysis and results in a less redundant latent representation. On the contrary, [R4] and [R5] are based on separable wavelets that adopt an isotropic scale for both directions.

   - Our FDWA can explore the cross-frequency correlations in both non-linear transforms and the entropy model. In contrast, [R4, R5] independently process each decomposed frequency component in the subsequent transform network, hindering the end-to-end optimized framework.

   - Our method ensures that **all features** are processed throughout the entire transform network and achieves multiscale and directional frequency decomposition **in each layer**, effectively reducing the spatial redundancy of each frequency component. In contrast, In [R4, R5], the high-frequency components extracted by the shallow layer are not further processed by subsequent layers, leading to spatial redundancies in the high-frequency subband. In contrast,

3. Compared with the handcrafted directional analysis tools in multiscale geometric analysis, we achieve **end-to-end optimization** to realize a novel anisotropic window attention.

[R1] Liu, Jinming, Heming Sun, and Jiro Katto. "Learned image compression with mixed transformer-cnn architectures." Proceedings of the IEEE/CVF Conference on Computer Vision and Pattern Recognition. 2023.

[R2] Zhu, Yinhao, Yang Yang, and Taco Cohen. "Transformer-based transform coding." International Conference on Learning Representations. 2022.

[R3] Zou, Renjie, Chunfeng Song, and Zhaoxiang Zhang. "The devil is in the details: Window-based attention for image compression." Proceedings of the IEEE/CVF conference on computer vision and pattern recognition. 2022.

[R4] Ma, Haichuan, et al. "iWave: CNN-based wavelet-like transform for image compression." IEEE Transactions on Multimedia 22.7 (2019): 1667-1679.

[R5] Ma, Haichuan, et al. "End-to-end optimized versatile image compression with wavelet-like transform." IEEE Transactions on Pattern Analysis and Machine Intelligence 44.3 (2020): 1247-1263.

---

### Meta-Review · Area_Chair_k5ia · 2023-12-10

**Metareview:**

The authors tackled the learned image compression problem with a transformer-based solution. They propose a frequency-aware module to perform multi-scale, directional analysis of a transformer features and employs FFT to weigh the frequency components. Improvements over prior works are reported on various datasets.

The major strengths are the state-of-the-art reported results and the novel architecture which does not significantly increases the complexity of prior state-of-the-art works.

The major weakness is the limited novelty, as all the components and ideas were introduced before. However, the overall architecture and solution is non-trivial novel.

The authors provided responses to all the reviewers' concerns and the reviewers are unanimously leaning towards acceptance of the paper (6,6,6,6).

The meta-reviewer after carefully reading the reviews, the discussions, and the paper, agrees with the reviewers and recommends acceptance.

**Justification For Why Not Higher Score:**

While the overall design and the performance of the method are strengths, the advances at the machine learning or theoretical level are limited and the interest for the addressed topic is rather limited within the ICLR community.

**Justification For Why Not Lower Score:**

Four reviewers and the meta-reviewer agree that the paper has merits, and the contributions are sufficient and of interest. There is no significant flaw to impede publication.

---

### Decision · Program_Chairs · 2024-01-16

Accept (poster)